# Primacy of vision shapes behavioral strategies and neural substrates of spatial navigation in marmoset hippocampus

Diego B. Piza [1,2], Benjamin W. Corrigan [1,2,3], Roberto A. Gulli [4], Sonia Do Carmo [5], A. Claudio Cuello [5], Lyle Muller [2,6] & Julio Martinez-Trujillo [1,2,7,8,9] ✉

The role of the hippocampus in spatial navigation has been primarily studied in nocturnal mammals, such as rats, that lack many adaptations for daylight vision. Here we demonstrate that during 3D navigation, the common marmoset, a new world primate adapted to daylight, predominantly uses rapid head-gaze shifts for visual exploration while remaining stationary. During active locomotion marmosets stabilize the head, in contrast to rats that use low-velocity head movements to scan the environment as they locomote. Pyramidal neurons in the marmoset hippocampus CA3/CA1 regions predominantly show mixed selectivity for 3D spatial view, head direction, and place. Exclusive place selectivity is scarce. Inhibitory interneurons are predominantly mixed selective for angular head velocity and translation speed. Finally, we found theta phase resetting of local field potential oscillations triggered by head-gaze shifts. Our findings indicate that marmosets adapted to their daylight ecological niche by modifying exploration/navigation strategies and their corresponding hippocampal specializations.

The hippocampus is a phylogenetically ancient structure of the mammalian brain that has been implicated in spatial memory and navigation[1–3]. Understanding how the hippocampus supports a cognitive map-like representation of the outer world first came from recordings of single-neuron spiking activity in freely moving rats, describing individual neurons that selectively increased their firing rate when subjects occupied a specific location within a maze[1,4]. Following the discovery of place cells, a rich diversity of spatial encoding neurons has been reported within the hippocampal formation and in functionally related brain areas[5–13]. Based on the results of spatial navigation studies in rodents, the hippocampus has been deemed a Global Positioning System (GPS) that enables the formation of a cognitive map of the environment, evidence supporting this has been consistently found in a multitude of conditions, like during microgravity[14] and in volumetric space[15]. However, recent navigation studies in primate species such as macaques and marmosets have not replicated the range of neuronal selectivities found in the hippocampus of rats and mice[16,17]. Thus, it is unclear whether the analogy of the hippocampus as a GPS generalizes to the aforementioned primate species.

In primates, hippocampus studies are much scarcer than in rats and mice. Some studies in macaque monkeys have reported that neurons in the hippocampus encode the direction of the subject's gaze in space (view)[16,18–25]. It has been proposed that diurnal primates' highly developed visual capabilities[26] may have shaped neuronal selectivities in the hippocampus[20,27,28]. Indeed, macaques and marmosets with

¹Schulich School of Medicine and Dentistry, Western University, London, ON, Canada. ²Robarts Research Institute, Western University, London, ON, Canada. ³Department of Biology, Faculty of Science, York University, Toronto, ON, Canada. ⁴Zuckerman Institute, Columbia University, New York, NY, USA. ⁵Department of Pharmacology and Therapeutics, McGill University, Montreal, QC, Canada. ⁶Department of Applied Mathematics, Western University, London, ON, Canada. ⁷Department of Physiology and Pharmacology, Western University, London, ON, Canada. ⁸Department of Psychiatry, Western University, London, ON, Canada. ⁹Department of Clinical Neurological Sciences, Western University, London, ON, Canada. ✉e-mail: julio.martinez@robarts.ca

diurnal lifestyle and foveal, stereoscopic color vision, have developed a head–gaze control system that allows orienting the fovea toward locations of interest while inspecting visual scenes[29–31]. However, several studies in non-human primates (NHPs) have mainly used paradigms in which the subject is placed in a primate chair with the head and/or body restrained while performing visual tasks on a computer screen[32–34]. These experimental paradigms may deprive the hippocampus of multisensory inputs such as vestibular and proprioceptive that occur during real-world navigation.

A few studies in rhesus macaques and squirrel monkeys[16,35–37] have reported that some hippocampal neurons are tuned for variables related to place, gaze direction or view, head movements and the interaction between view and place. One study trained marmosets to navigate a linear maze under constrained conditions (tethered) and described the existence of place-like cells in the hippocampus[17]. Interestingly, they describe that rhythmic hippocampal theta oscillations are not omnipresent during locomotion-exploratory behavior[17], as is the case in mice and rats[3,38]. In other mammalian species, such as the Egyptian fruit bat[39] theta oscillations in the hippocampus occur in short bouts. Finally, some studies have documented that in humans and macaque monkeys, theta oscillations are coupled to saccades, and they have variable frequencies[40–42].

In the present study, we test the hypothesis that diurnal primates have developed different exploration–navigation strategies compared to nocturnal rodents such as rats, and that such strategies have shaped the physiology of the hippocampus. We investigate the exploration–navigation strategies of marmosets during unrestricted 3D foraging and compare them to those of freely moving rats. We built a setup that allowed for continuous tracking of the marmoset body position and head direction in 3D and recording neural activity from the hippocampus (CA3 and CA1) wirelessly. We found that marmosets navigate 3D environments using quadrupedal locomotion, during which the head–gaze remains relatively stable. They make frequent stops, during which they execute sequences of rapid head–gaze movements toward locations of interest (visual exploration). This is different from rats that often move their heads at low velocities during locomotion, "scanning" the environment with their whiskers[43]. Additionally, we use wireless recordings of neural activity and demonstrate that marmoset hippocampal neurons in subfields CA3 and CA1 predominantly encode a mix of variables related to 3D head/gaze direction, speed and position in space. We found a predominance of 3D view, head direction and spatial position coding in putative pyramidal (principal) cells, and a mix of 3D angular head velocity (AHV) and translation speed (TS) in putative interneurons. We demonstrate that from small ensembles of mixed selective neurons, weakly tuned for spatial position, we could reliably decode the subject's position in the maze. Finally, we demonstrated that rapid head–gaze movements reset local field potentials (LFP) theta oscillations in the hippocampus, which coincides with an increase in the response of interneurons followed by modulation of pyramidal neurons' firing.

## Results

### Spatial exploration strategies in marmosets and rats

We measured position and head direction signals in two freely moving marmosets (one male and one female) while they foraged through a 3D maze searching for reward (e.g., condensed milkshake or marshmallows; Fig. 1a; Figs. S1g, S2a; "Methods"). A reward was delivered at 1 of the 12 possible maze locations randomly selected (see below), by either cueing the subject with an LED light or waving a hand at the reward location; this reward schedule was used to encourage navigation of the 3D maze. Recorded data consisted of six different variables, three for the position of the subject in the maze (horizontal or $X_p$, depth or $Y_p$, vertical or $Z_p$), and three for rotational head direction ($X_h$, $Y_h$, $Z_h$) yaw, roll and pitch. Body position signals were low-pass filtered at 4 Hz. TS was calculated from the position signals (see "Methods").

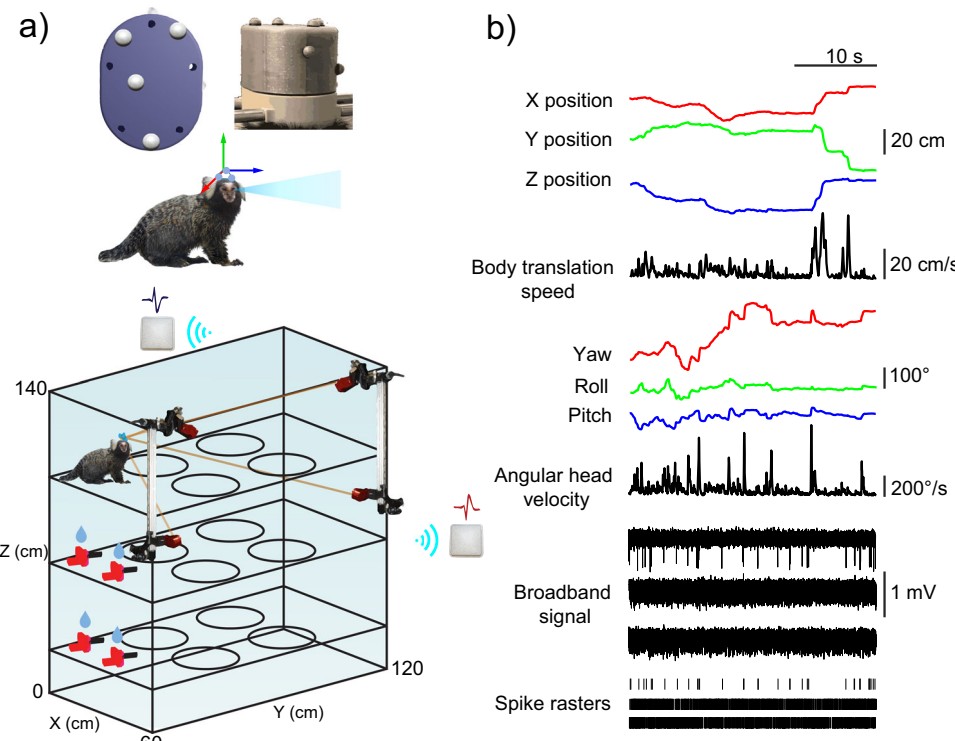

**Fig. 1 | Behavioral setup and data traces. a** Top: recording chamber cap cover with a schematic of marker positions that allow tracking position and direction of the subject's head. Bottom: a schematic of the maze and wireless recording setup.

**b** Data traces with three example single cells. The type of signal and corresponding units are indicated. In the spike rasters, each vertical line indicates an action potential.

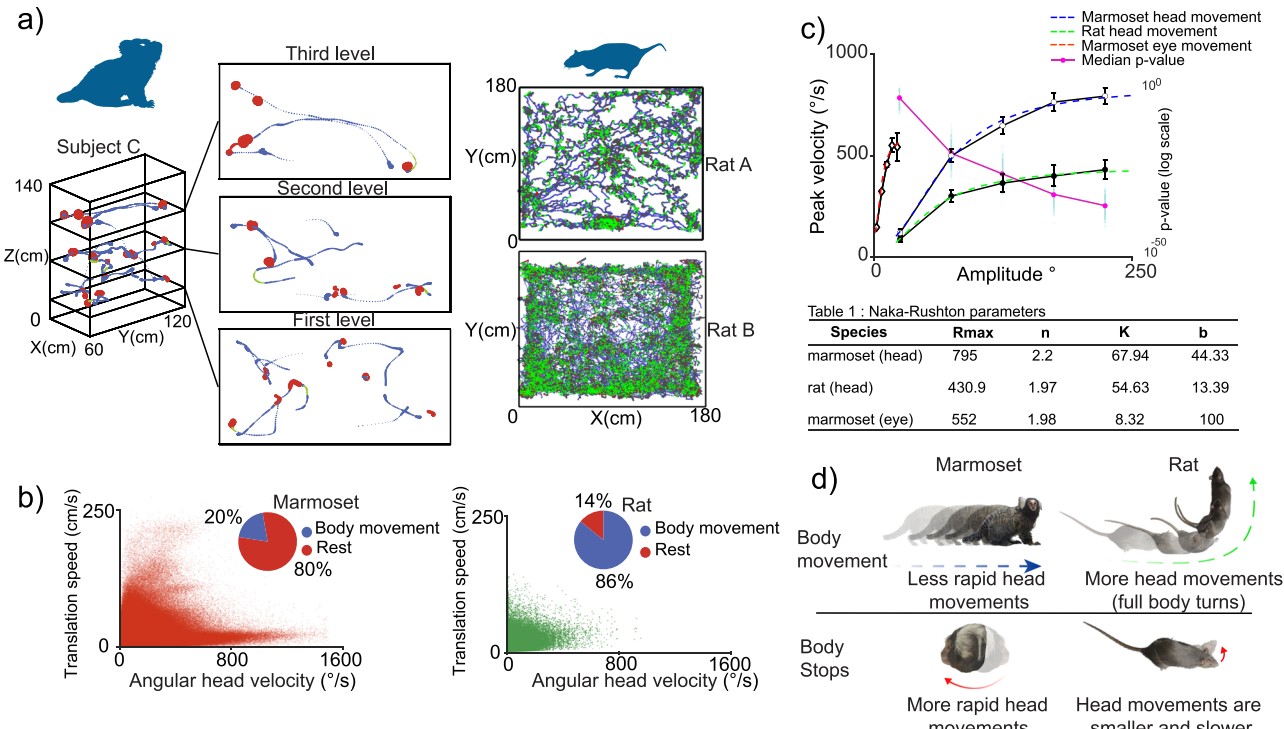

**Fig. 2 | Exploration strategies in marmosets and rats. a** Selected trajectory examples of Subject C (left) and two different rat subjects (right) during foraging (rat Subject A on the top panel and rat Subject B on the bottom panel). The size of the marker is proportional to the time spent at that location, Red = rapid head movement, Blue = body translation movement and Green = both head and body movement were present. **b** Distribution of AHV vs TS, pie chart displays the percentage of head movements that happen during body translation movement or rest (defined as no body movement present), for both marmoset (left) and rat (right).

**c** Mean values for the main sequence of head movements for both marmoset (head and eye movements) and rat (head movements), error bars correspond to 95% confidence interval. A Naka–Rushton function was fit to the mean values (dashed line), two-sided Wilcoxon signed-rank test bootstrapped *p* values (marmoset head movement data were re-sampled to match rat head movement counts, this process was repeated 500 times) are displayed on a log-scale (cyan dots) and median *p* values are shown as pink line and dots. **d** Schematic of the two different exploration strategies observed in marmosets and rats during foraging.

$X_h$, $Y_h$, and $Z_h$ rotation vectors were used to calculate the AHV (see "Methods"). We then classified rapid translation or rapid head rotation movements based on a minimum speed/velocity threshold (higher than 15 cm/s TS and 200°/s AHV, for body translation and head rotation movements, respectively) and amplitude threshold (30 cm and 16° for body translation and head rotation movements, respectively).

We observed that during foraging marmosets alternate between periods in which the subjects translate in 3D space (quadrupedal locomotion) while the head remains "fixed" relative to the body, and periods in which they stop translating in space (body stationary), and the head frequently rotates relative to the body (Supplementary Movie 1). During the translation periods, subjects were observed to travel most paths using multidirectional trajectories (multiple heading directions), spanning all quadrants (Fig. S3a, b). During the body stationary periods, they explore the environment through frequent rapid head movements that end in gaze fixations[44], resembling the way humans and other primates freely explore visual scenes[30]. We will refer to these rapid head movements as head–gaze shifts since the main goal is to align the gaze with objects/locations of interest. To corroborate this observation, we computed translation trajectories and associated velocities. Indeed, marmosets alternated between translations during which head–gaze movements were scarce (Fig. 2a, left, blue trajectories) and stops during which head movements were frequent (Fig. 2a, left, red dots).

To contrast the navigation strategies of marmosets and rats, we performed the same analyses in a dataset available online (CRCNS.org, Data sharing, hc-2 data set) collected while rats were foraging in a maze searching for rewards after being habituated for at least 3 days to the arena. Rats were video-tracked (sampled at 39.06 Hz) using two LEDs

of different colors, placed on the front and back of the head, from which the position and head direction can be measured[45]. Rats are nocturnal; their vision has adapted to detect motion in dim light conditions. Their retinas possess ultraviolet-sensitive cones, and the majority of their optic nerve axons target the superior colliculus (SC) rather than the lateral geniculate nucleus[46]. Although rats possess visual capabilities and can perform visual tasks[47], they have no well-defined fovea, their spectral sensitivity is considerably smaller than that of diurnal primates, and the anatomical position of their eyes does not enable stereo vision to the same degree as front-positioned eyes primates[46,48]. Consequently, rats must rely to a larger degree on olfaction and whisking to sense the environment. Most of the rat's eye movements are often disconjugated[49] and stabilize the eyes against movements of the head.

We found that rats' head movements often occurred simultaneously with the body translation movements during locomotion (Fig. 2a, right, green dots). In marmosets, 80% (99% confidence interval [79%, 80%]) of the head movements occurred during body stops while in the rat only 14% (99%, confidence interval [12%, 14%]) of the head movements occurred during body stops (Fig. 2b, left pie chart). On the other hand, in marmosets, only 20% (99% confidence interval [19%, 20%]) of the head movements occurred during body translations while in rats 86% (99% confidence interval [85%, 87%]) of the head movements occurred during translations (Fig. 2b, right pie chart). The range of TS and AHV in the marmoset appears larger than in the rat which may reflect different adaptations in the two species. The AHV in the marmoset has a long tail along the *X*-axis corresponding to low TS compared to the rat (Fig. 2b). This is due to the high frequency of rapid head–gaze movements in the marmoset when the body is stationary.

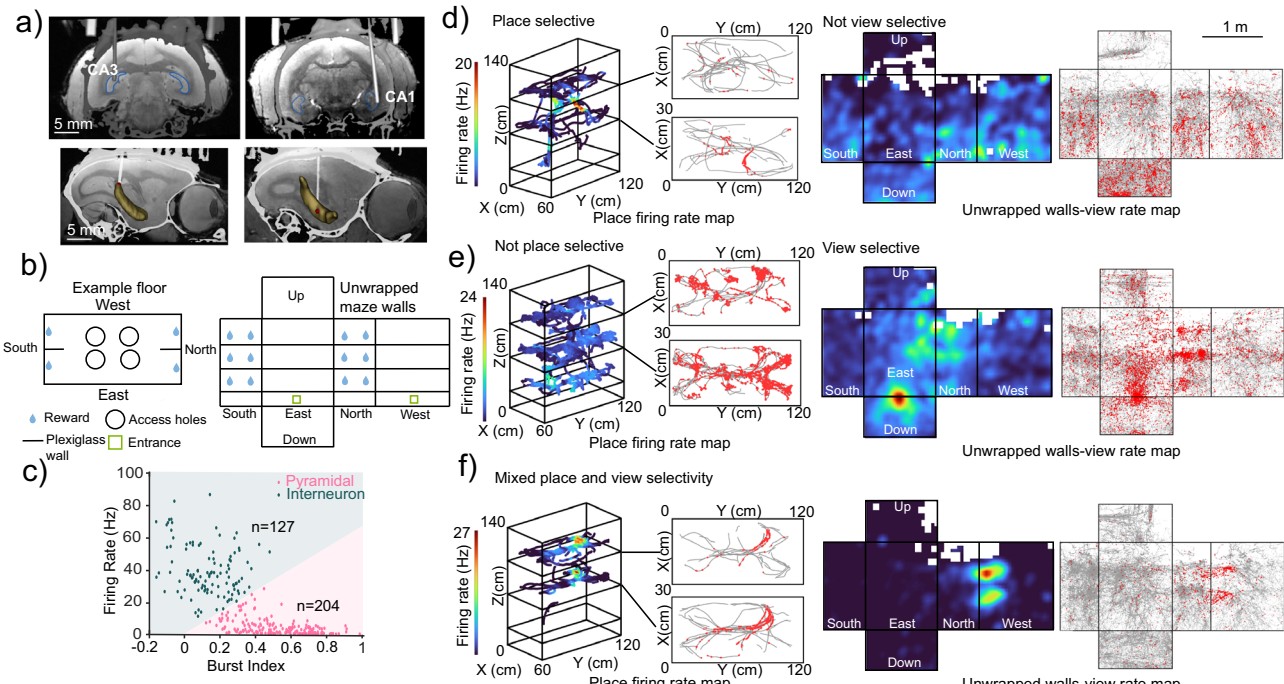

**Fig. 3 | Single neuron responses to view and place. a** Top: MRI and CT imaging showing the final electrode location. Bottom: 3D reconstruction of the segmented hippocampus showing electrode location for Subject P (left) and Subject C (right). **b** Schematic representation of the 3D maze. **c** Electrophysiological parameters used to classify cells into putative pyramidal cells ($n = 204$) or putative interneurons ($n = 127$), the cluster boundaries are calculated using the $k$-means algorithm. **d** Single-cell example of a place cell. Firing rate maps for both places (left panels, 3D trajectories colored with firing rate, black lines coming off the 3D maze indicate a flat top view of the top two floors, gray lines are trajectories and red dots are spikes) and view (right panels, unwrapped walls of the maze, color indicates firing rate when view is projected at that location, on the right gray lines correspond to view trajectories and red dots to spikes). The color indicates the firing rate. **e** Single-cell example of a view cell (similar panel configuration as in (**d**)). **f** Single-cell example of a mixed place and view cell (similar panel configuration as in (**d**)).

Rapid head–gaze movements in the marmoset have the signature of gaze shifts composed of coordinated movements of the eyes and the head. The main sequence, described as the relationship between the amplitude and peak velocity of eye saccades has been used to characterize gaze shifts[50]. During gaze shifts the peak velocity/speed increases monotonically with increases in head movement or saccade amplitude, progressively saturating for large movements[30]. Marmoset rapid head movements, much like eye movements, are stereotyped ballistic movements used for visual exploration; and the main sequence has been used to describe their kinematics[44]. We hypothesize that in the marmoset the main sequence of the head movements will follow a similar profile as the main sequence of gaze shifts in macaques. On the other hand, rats, lacking a sophisticated head–gaze apparatus would systematically show lower peak velocities relative to marmosets.

To test this hypothesis, we computed the main sequence of head movements (movement amplitude vs peak velocity) in both species. In both, marmosets and rats head peak velocity increased as a function of movement amplitude (Fig. 2c) indicating a general kinematic principle also reported in other species[51]. Importantly, head movements of the same amplitude have higher mean peak velocity in the marmoset than in the rat (two-sided Wilcoxon signed-rank test, $p < 5 \times 10^{-324}$, $Z = 113.8$). The maximum response ($R_{max}$) of a Naka–Rushton function fit to the velocity as a function of movement amplitude is higher for marmosets compared to rats (795°/s vs 430.9°/s; Fig. 2c, Fig. S1c, d). These results are consistent with marmosets' use of head–gaze shifts to visually explore the environment during body stops to position and stabilize the fovea on objects of interest. On the other hand, rats use head scanning mainly during locomotion to orient the whiskers and facilitate exploration[52,53]. Rats may also use vision but with their lack of

fovea and poor color and stereo vision, they may not require image stabilization to the same degree as marmosets do[49,54].

We did not measure eye position in the marmosets in the 3D maze, but instead relied on head movements as a proxy for gaze direction, thus marmosets may have moved the eyes-in-head during exploration producing misalignments of head and gaze. We have previously demonstrated in macaques that after a head–eye gaze shift the eyes and head tend to be aligned at the final position[51]. However, we tested marmosets during several sessions in which they were seated on a primate chair, and we fixed the head while they explored a visual scene on a computer monitor. We measured eye position and computed the amplitude and peak velocity of eye-in-head saccades. Interestingly, we observed that saccades were in their majority shorter than 5° (Fig. 2c; Fig. S1e, f), which coincides with the lower limit of the head movement distribution. So, if we were to estimate the gaze position from the head direction in the marmoset, the error could be as large as 5°. Since our goal was not to obtain an accurate estimate of gaze position but to compare marmosets and rats' visual exploration strategies, we conclude our measurements have sufficient resolution to justify our main conclusions.

## Coding of space by single neurons in the freely moving marmoset hippocampus

We recorded the responses of single neurons and LFPs in the hippocampus of two common marmosets by implanting chronic microwire brush arrays (MBA, Microprobes for Life Science, Gaithersburg, MD) in the subfields CA3 (Subject P) and CA1 (Subject C). We verified the final position of the electrodes using 9.4T MRI and micro-CT imagining co-registration (Fig. 3a; Fig. S2b; "Methods"). Data acquisition was accomplished using wireless telemetry (CerePlex Exilis, Blackrock

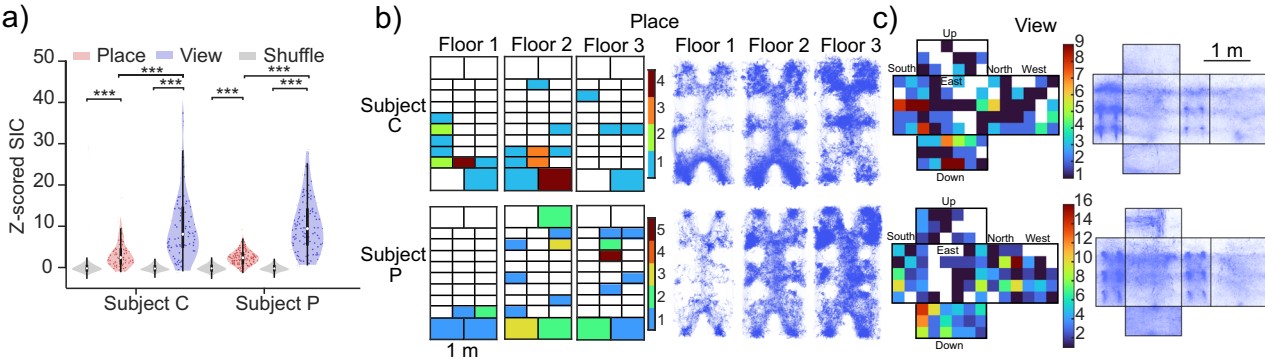

**Fig. 4 | SIC comparison between view and place and place/view field locations.**
**a** Violin[169] distribution of Z-scored SIC for putative pyramidal cells (n = 204) gray
distribution is shuffled control. **b** Left: place field location for all place cells and
separated according to the three distinct floors of the 3D maze, color indicates the
number of place fields at that location, white indicates bins without place fields.
Right: map of all spatial trajectories traveled across (n = 26 and n = 33 sessions) for
Subject C (top) and Subject P (bottom), respectively. Color intensity reflects more
time spent in that position. **c** View field locations for all view cells, color indicates
the number of view fields at that location, white indicates bins without view fields.
Right: map of all view-gaze positions (n = 26 and n = 33 sessions) for Subject C (top)
and Subject P (bottom), respectively. Color intensity reflects higher counts.

Microsystems, Salt Lake City, UT) while the marmosets foraged for
rewards at reward ports of a 3D maze (Fig. 3b; "Methods"). We defined
view as the facing location of the marmoset at any given point in time
(linear projection of the head direction onto the walls of the maze, see
"Methods")[16]. We recorded a total of 331 neurons in both subjects (178
in Subject C and 153 in Subject P).

Excitatory pyramidal neurons are commonly regarded as the cells
responsible for transmitting and processing spatial information in the
hippocampus[1,55–57], while inhibitory interneurons have been associated
with the encoding of non-spatial variables like speed[58], or with syn-
chronizing activity in the hippocampal network[59]. To quantify spatial
tuning to either place or view with conventional methods, we classified
neurons based on their bursting properties and firing rate[60–64] into
either putative interneurons or putative pyramidal cells (Fig. 3c), and
we used exclusively putative pyramidal cells to assess variables related
to spatial tuning. A total of 204 cells were classified as putative pyr-
amidal and 127 were classified as putative interneurons.

Some putative pyramidal cells showed selectivity for the subject's
position in the maze while they were poorly selective for view (exam-
ple in Fig. 3d). Other cells show view selectivity but poor place selec-
tivity (example in Fig. 3e). Other cells show a mix of view and place
selectivity (example in Fig. 3f). For the analyses of spatial selectivity, we
calculated the spatial information content (SIC)[65–67] of putative pyr-
amidal cells. The SIC quantifies the amount of information (bits) each
neuronal spike transmits about the location of the subject. SIC was
computed for each putative pyramidal neuron, and an SIC shuffled
control was computed by circularly shifting spike times by a random
duration a total of 5000 times. Neurons with SIC that exceeded the
95th percentile of the null distribution were classified as selective.
Overall, more neurons were classified as view cells than place cells
(Fig. 4a). We found that 66 (32%) of cells were place selective (Fig. 4b)
and 159 (76%) were view selective (Fig. 4c). For Subject C, out of all the
sampled view bins (106 total), 76.4% (81/106) (99% confidence interval
[64.5%, 88.4%]) had at least one cell that exhibited selectivity for that
bin. Similarly, for Subject P 78.3% (83/106) (99% confidence interval
[68.6%, 88%%]) had at least one selective view field. In contrast, of all
visited place bins (84 total), in both subjects, 25% (21/84) (Subject C
99% confidence interval [12.11%, 36.1%], Subject P 99% confidence
interval [14.1%, 34.2%]) had at least one selective place field.

## Coding of speed by single neurons in the hippocampus of freely moving marmosets

Neurons in the rodent hippocampus show selectivity for the TS of the
animal locomoting in a maze or running in a wheel[11,12,58,68–73].

However, little is known about the presence of speed cells in the hip-
pocampus of NHPs. A previous study[16] reported encoding of both TS
and AHV in freely moving macaques, mainly in putative interneurons.
We found no reports in the common marmoset.

For this analysis, we included both putative interneurons and
pyramidal cells. We found single neurons that vary their response rate
as a function of AHV (Fig. 5a, left) and TS (Fig. 5a, right). To quantify
speed encoding we calculated the speed score as defined in refs. 58,74
(Pearson correlation coefficient between the time series of firing rate
and either AHV or TS). To select the neurons that were responsive to
either AHV or TS, any given cell had to meet these two criteria: (1) as
reported by a previous study[58], a speed score higher than 0.3, and (2)
the cell's speed score had to be higher than the 95th percentile of
speed scores in a shuffle null distribution (1000 permutations calcu-
lated by circularly shifting the vectorized spike raster relative to the
equally sized speed vector) (Fig. 5b; Fig. S4f–i). We found a total of 43
(33%) speed-selective cells.

A significant amount of the speed cells encoded both, TS and AHV
(Fig. 5c, 20 AHV cells, 4 TS cells, 19 mixed cells). Interestingly, neurons
encoding AHV often show selectivity for only that variable, but most
neurons encoding TS also showed selectivity for AHV (AHV cells 90%
confidence interval [34%, 59%]; TS 90% confidence interval [1%, 13%]).
As a control we investigated whether speed cells were predominantly
putative interneurons (Fig. 5d, e; Fig. S4j, k, proportion of speed cells
that were putative pyramidal cells 90% confidence interval [3%, 18%];
proportion of speed cells that were putative interneurons 90% con-
fidence interval [82%, 97%]), furthermore, the median AHV and TS
score for the population of putative interneurons was higher than in
putative pyramidal cells (two-sided Wilcoxon signed-rank test, TS
$p = 3 \times 10^{-22}$, $Z = 9.7$; AHV $p = 1.3 \times 10^{-25}$, $Z = 10.5$) corroborating that, as
in the rat, speed tuned cells are mainly interneurons[58].

## Mixed selectivity in the marmoset hippocampus

To quantify mixed selectivity for the different variables (space, view,
head direction and speed) in the recorded neurons we use a general-
ized additive model[75] (GAM) that fits the neuron's response to beha-
vioral parameters described as the sum of fitted, cross-validated spline
functions. We trained two different model types, one for putative
pyramidal cells and the other for putative interneurons. The output
model can have either none, one (first-order model) or multiple vari-
ables (i.e., second- or third-order models) as significant predictors of
the firing rate of single neurons, we can then classify cells that encode
multiple variables as cells with mixed selectivity. For putative pyr-
amidal cells, the model included place, view, and head direction as

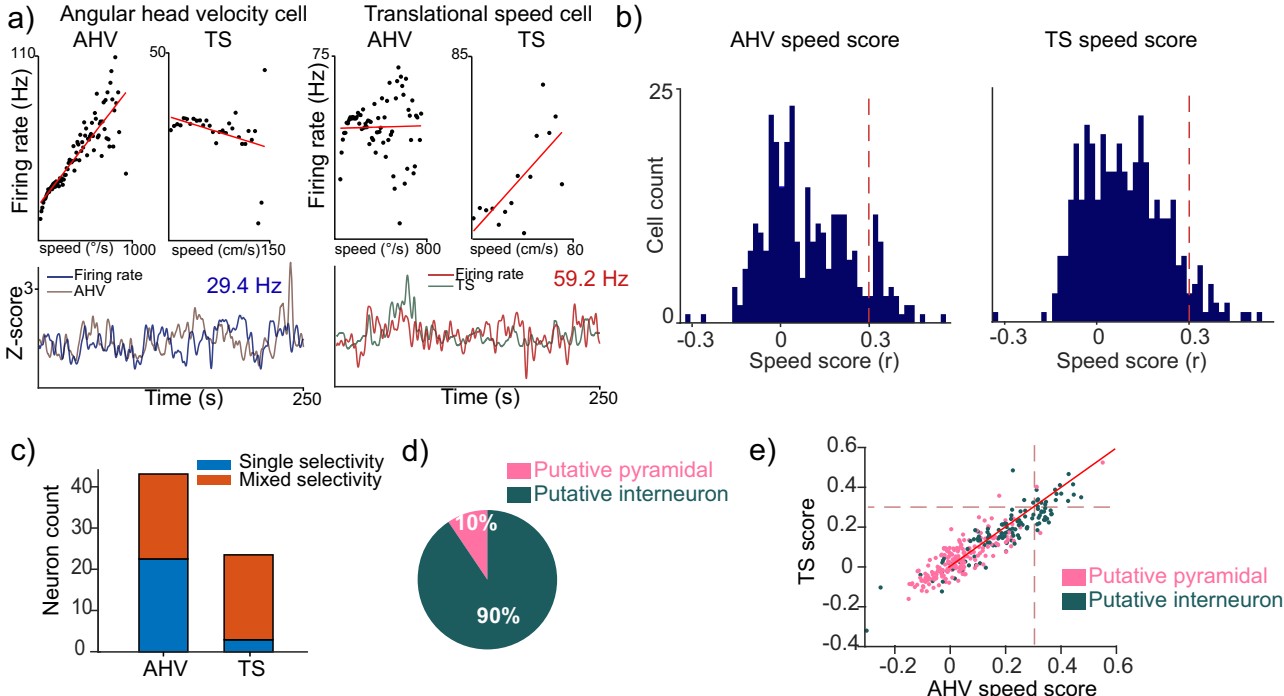

**Fig. 5 | Single neuron responses to TS and AHV. a** Example cells. On the left for AHV, the cell increases the firing as a function of AHV but not TS. Right: TS cell firing increases as a function of TS but not AHV. **b** Distribution of speed score (Pearson correlation between time series of firing rate and speed) for both TS and AHV for all cells, the red dotted line at a 0.3-speed score, indicates the threshold to define significant encoding (bin width = 0.017). **c** AHV and TS significant cells count distribution, orange color indicates cells that are both AHV and TS significant cells. **d** Distribution of speed cells according to putative cell type. **e** Speed score distribution for both AHV and TS labeled according to cell type, (green = putative interneuron, pink = putative pyramidal), the red dotted line indicates a speed score of 0.3. Solid red line is a line of slope = 1, corresponding dots below it indicate higher AHV than TS, and dots above it indicate the opposite.

predictors (Fig. 6a). The putative interneuron model included AHV and TS as predictors (Fig. 6c). Across all cell types, 174/331, 52.6% significantly encoded at least one variable (putative pyramidal cells 73/204, 35.8%; putative interneurons 101/127, 79.5%); amongst all encoding cells, mixed selectivity was predominant over single selectivity (162/174, 93.1% mixed selective; putative pyramidal cells: 67/73, 91.8% mixed selective; putative interneurons: 95/101, 94.1% mixed selective).

In the putative pyramidal model (Fig. 6b), place was exclusively encoded in combination with either view or head direction, there were no cells whose firing rate encoded only place, with the large majority of cells significantly encoding the three variables (view: 3/204, head direction: 3/204, view + head direction: 5/204, view + place: 4/204, place + head direction: 5/204, view + place + head direction: 53/204, Fig. S6 for rate map examples, Fig. S7a, b for individual marmoset results). For the putative interneuron model (Fig. 6d; Fig. S7c, d), we found 3/127 (2.4%) AHV cells, 3/127 (2.4%) TS speed cells and 95/127 (74.8%) mixed selective cells (AHV + TS). In summary, for the putative pyramidal model, single behavior selectivity is dominated by visuospatial variables (view + head direction) rather than place alone. For the putative interneuron model, selectivity was similar for TS and AHV. Finally, for both putative pyramidal and interneuron cells, mixed selectivity for combinations of different variables was predominant.

### Ensembles of mixed selective neurons encode place

Our previous results demonstrate that individual cells within the marmoset hippocampus are weakly selective for place, and rather encode a combination of variables, with a higher proportion of cells encoding view or head direction relative to place. This observation may suggest that the amount of place-related information in the marmoset hippocampus is relatively limited. However, previous research has demonstrated that populations of neurons, referred to as ensembles, can still contain substantial amounts of information about task-related variables, even if individual cells are poorly tuned[32,76,77]. Furthermore, numerous studies, conducted primarily in rodents have amassed a wealth of evidence demonstrating that the subject's position can be decoded from the firing rate of hippocampal cell ensembles[69,78,79] achieving accuracies of ~10 cm (approximately equivalent to the subjects' body size)[80]. To further explore this issue, we used firing rates of non-simultaneously recorded pseudo-population of mixed selective putative pyramidal cells in areas CA3 and CA1 to decode the animals' location in the maze.

We first divided the maze into two spatial regions (bins) per floor (Fig. 7a). From the six resultant bins, we identified the four bins that best-optimized visitation frequency and neuron counts across all sessions for each subject, (Fig. 7a, d, top). Using firing rates obtained during periods where the subject's head velocity was low (<200°/s, see "Methods"), we decoded the subject's location employing a linear multi-class support vector machine (SVM) classifier. To identify the combination of neurons (ensemble) that provides the best decoding accuracy, we used an "ensemble construction" method previously described[81,82]. We iteratively tested combinations of different neurons from a pool of a pseudo-population of putative pyramidal neurons. In the first iteration, we train one classifier per neuron, and the neuron exhibiting the highest decoding accuracy is then selected as the "ensemble seed", for the second iteration, we train all the possible two-neuron combinations that include the previously selected units along with the remaining units in the pseudo-population; then the best two-neuron ensemble is selected. This process continued with each subsequent iteration, involving the training of an $n + 1$ ensemble and the selection of the "best ensemble", until performance becomes asymptotic (e.g., at $n = 20$ cells). We constructed two different ensembles using this method, one where the pool of neurons included all recorded putative pyramidal cells, we called this the "all units best ensemble" (pool size = Subject C 75 cells, Subject P 119 cells), and a second

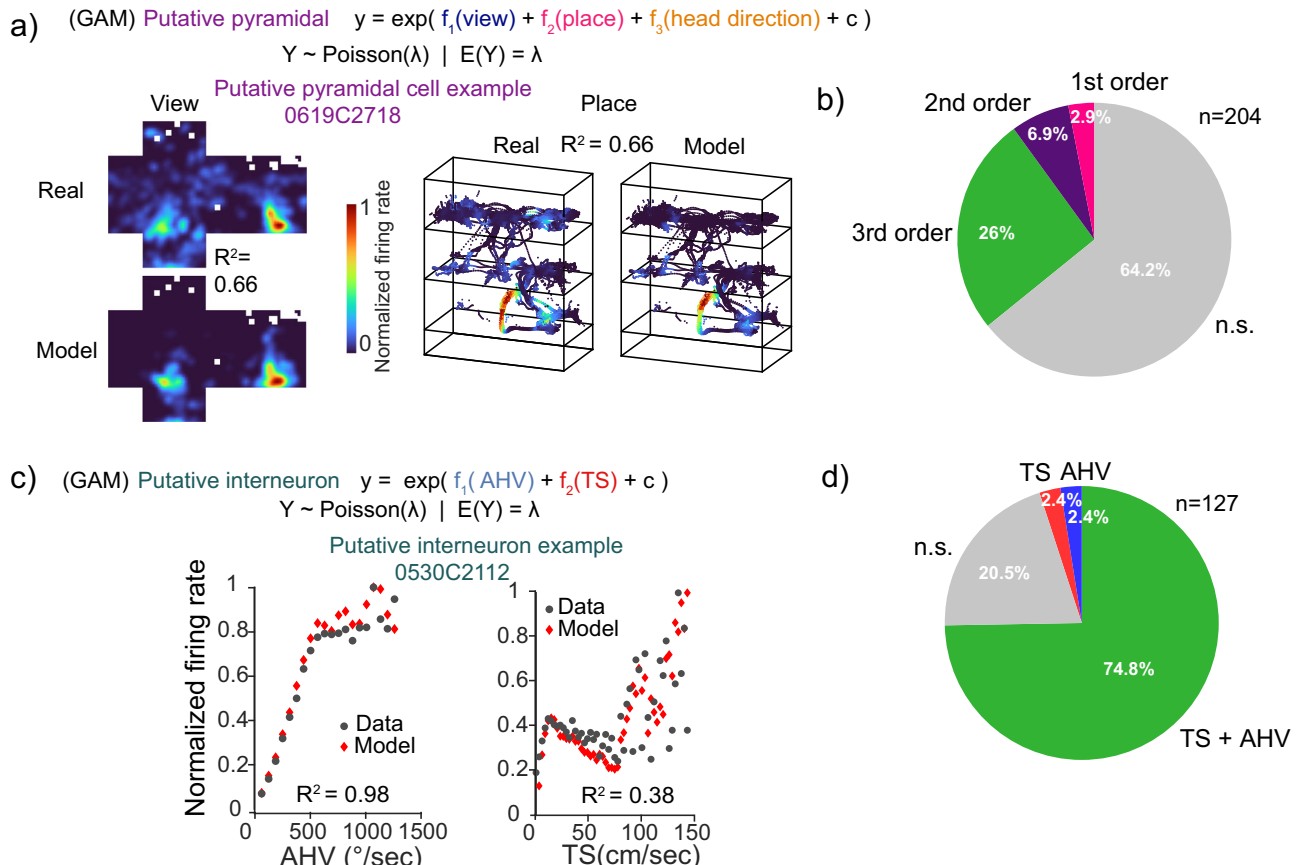

**Fig. 6 | Mixed selectivity GAM encoding model.** A cross-validated GAM model was fitted to **a** putative pyramidal cells (view + place + head direction (h.d.)), single-cell model fit example for view (left) and place (right), rate maps correlation shown as $R^2$. **b** Proportion of encoding putative pyramidal cells (pie chart). **c** GAM fitted to putative interneurons (AHV + TS), single-cell model fit example for AHV (left) and TS (right), the goodness of fit is shown as $R^2$. **d** Proportion of putative interneuron encoding cells (pie chart).

ensemble where the cells that significantly encoded at least one spatial variable, as per the previous GAM analysis, were excluded from the pool of neurons, we called this the "no significant units best ensemble" (pool size = Subject C 61 cells, Subject P 78 cells). For both pools of neurons, we only included cells from sessions where subjects sampled each spatial bin at least 50 times. The latter was done to have sufficient data to train the classifiers. We fit a Naka–Rushton function to the decoder performance as a function of ensemble size and calculated statistics on the fit coefficients.

We decode place significantly above chance level (1/4, 0.25) "all units best ensemble" decoding accuracy (Fig. 7a, d, blue line; Fig. 7c, f), Subject C = 0.45 (95% confidence interval [0.44, 0.46]); Subject P = 0.49 (95% confidence interval [0.48, 0.5]). The "all units best ensemble" decoding accuracy was significantly higher than the "no significant units best ensemble" decoding accuracy (Fig. 7a, d, pink line), Subject C = 0.42 (95% confidence interval [0.41, 0.43]); Subject P = 0.42 (95% confidence interval [0.41, 0.43]). The latter indicates that neurons with significant fits for the GAM model contributed substantially to decoding accuracy. However, the fact that ensembles of neurons without significant fits also produce decoding accuracies higher than chance suggests that many of these neurons contain information about place. We further tested whether our ensemble construction method was effective in selecting combinations of the most informative neurons from the pseudo-population by training a decoder on "random combination ensembles". We randomly generated 100 different and unique combinations of 20 cells and decoded place from those ensembles (Fig. 7a, d, cyan line). The "random combination ensemble" decoding accuracy was significantly lower

than the one of optimized ensembles, Subject C = 0.35 (95% confidence interval [0.35, 0.36]); Subject P = 0.34 (95% confidence interval [0.34, 0.35]).

Finally, we examined the encoding profiles of the neurons comprising the "all units best ensemble". Specifically, we focused on identifying the best-encoded spatial variable based on the GAM model. We found that the cells part of the ensemble encoded a mixture of view, head direction and place (Fig. 7b, e). Interestingly, despite place being the decoded variable, there is a predominance of view and head direction as the single cells' best-encoded variables in the "optimized" ensembles. The latter demonstrates that spatial position could be decoded from our population of mixed selective neurons. Furthermore, it suggests that view and head direction can provide information about place and may explain the exploratory gaze behavior marmosets exhibit when stationary.

**Marmoset rapid head movements trigger LFP theta phase-resetting and modulation of single-cell responses**

Rhythmic, LFP theta oscillations in the hippocampus have been widely documented in rodents such as rats and mice[3,38]. Theta oscillations are thought to coordinate hippocampal cell assembles, and it has been proposed that they are involved in memory formation[83–85]. However, in some species like humans, macaques, marmosets and bats, theta oscillations are not present as a rhythm during locomotion but appear in short bouts[16,17,86–88]. In humans and macaques, saccadic eye movements are thought to reset the phase of theta oscillations[40–42,89,90]. Here we hypothesize that head–gaze movements in the marmoset will induce theta phase resetting, which modulates neuronal activity

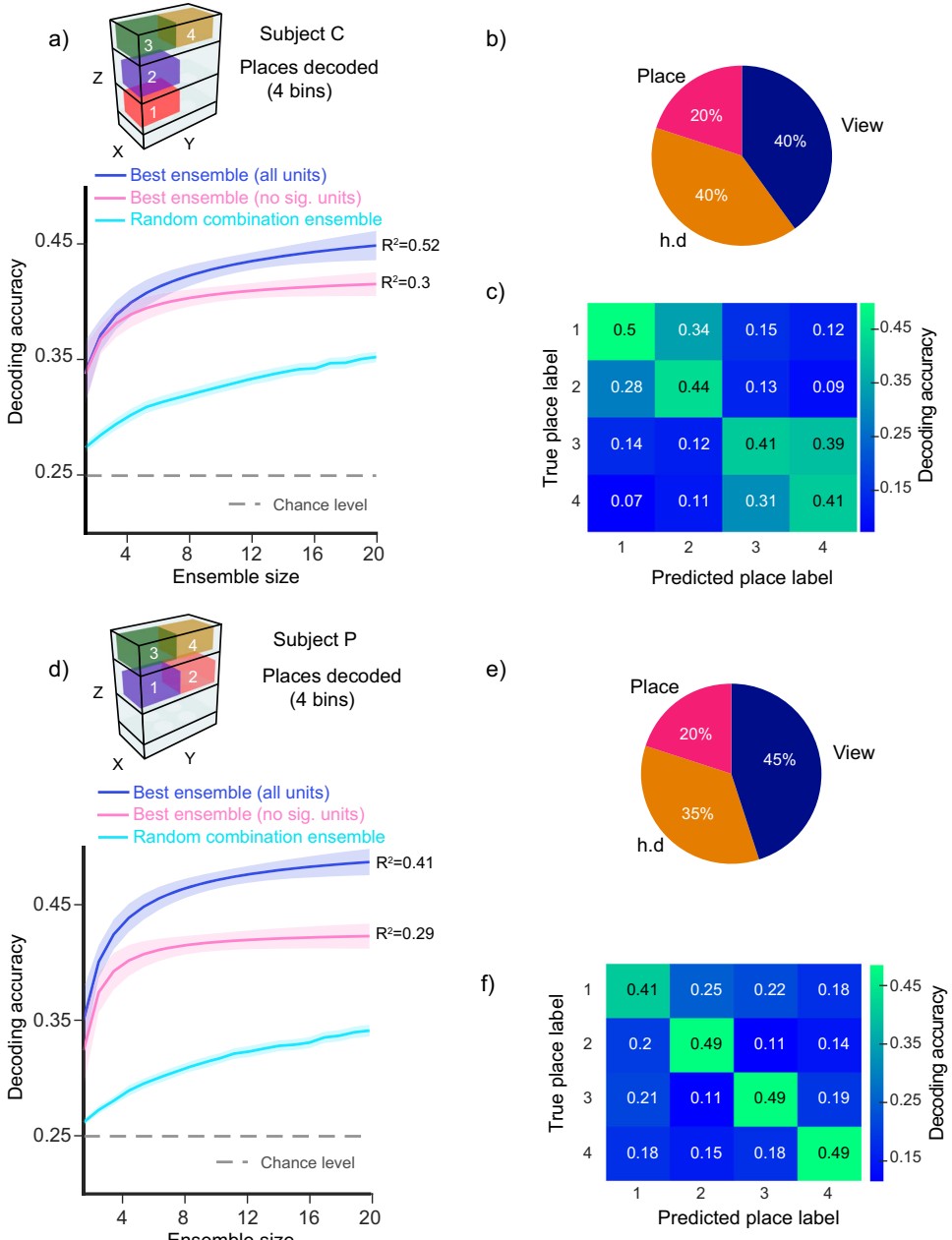

**Fig. 7 | Population decoding of place using a linear SVM classifier. a, d** Top: 3D diagram of the binned place locations used to decode the subject's position, bottom: blue and pink lines correspond to a Naka–Rushton function fit to the mean decoding accuracy (*y*-axis) as a function of ensemble size (number of neurons, *x*-axis), shaded area corresponds to 95% confidence intervals. Blue solid lines correspond to the best ensemble constructed from a pool of all recorded putative pyramidal neurons. Pink solid lines correspond to the best ensemble constructed from a pool of non significantly selective cells (as per GAM encoding analysis). $R^2$ goodness of fit value is reported. The cyan lines correspond to the mean decoding accuracy of a randomized combination of neurons (100 iterations), shaded area corresponds to 95% confidence intervals. The gray dashed lines correspond to chance decoding accuracy (1/4, 0.25). **b, e** Best-encoded variable proportion (as per GAM encoding analyses) for the combination of *n* = 20 neurons part of the best ensemble pooled from all the single units (**a, d**; blue line). **c, f** Confusion matrix derived from the best ensemble classification accuracy (**a, d**; blue line).

during visual exploration (when the animals acquire information about the environment) aiding memory encoding.

We aligned the LFPs to the peak velocity of head movements (see "Methods") and computed average LFPs (Fig. 8a). We also aligned spectrograms corresponding to LFP traces and computed their averages. We found the strongest power in the 4–15 Hz band with a peak at the theta frequencies (4–10 Hz) around the onset of the head movement (Fig. 8a). We observed an increase of theta phase alignment (resetting) (Fig. S8a, b), along with an increase in Rayleigh test of uniformity *Z* values during the peri-head movement periods (Fig. S8c).

Critical *Z* values were significant (*p* < 0.01) starting at -100 ms prior to head movement peak velocity. To determine whether head–gaze theta phase resetting was accompanied by a modulation of neuronal firing we obtained the average firing per single cell as a function of head movement events. We found that in both putative interneurons and pyramidal cells, spiking activity was modulated by the occurrence of a head movement (Fig. 7b; Fig. S9a–c). To quantify the precise nature of this modulation, we calculated a firing rate shuffled distribution through circular permutation of spike times, repeating the process 1000 times. Significance was evaluated using eight different bins, each

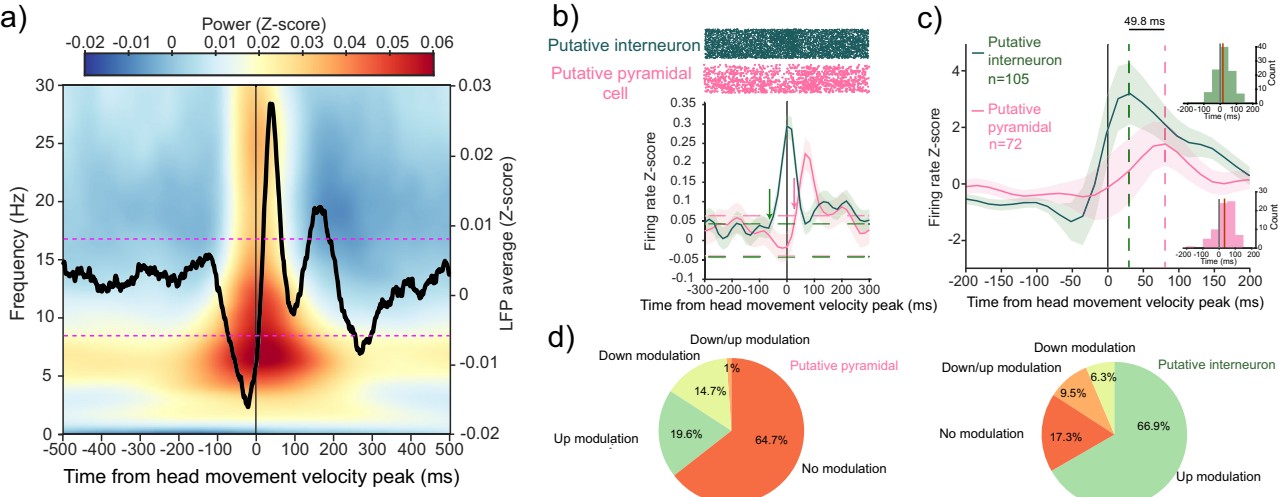

**Fig. 8 | Theta oscillations and single neuron phase resetting to head movements. a** Head–gaze phase resetting of the average LFP aligned to head movement peak velocity (0 point). The color scale indicates average time–frequency representation (TFR) as Z-scored LFP power. Pink dotted lines indicate 95% confidence interval of the mean LFP signal at time −200 ms. **b** Single-cell mean firing rate aligned to head movement peak velocity for a putative pyramidal (pink) and a putative interneuron (green) cell. Shaded area corresponds to 95% confidence interval of the mean firing and dotted lines represent 95% confidence intervals of the shuffle null distribution, arrows indicate time points when the firing rate becomes significantly modulated against that threshold. **c** Average Z-scored firing rate phase resetting for all head movement-modulated putative pyramidal (pink, n = 72) and putative interneuron (green, n = 105) cells; shaded area corresponds to 95% confidence interval and the dotted line indicates the time of maximal activation. Right histograms represent the distribution of times when the firing rate becomes significantly modulated (bin width = 50 ms) for all modulated cells. Dark gray continuous line indicates 0 point, red continuous line on the histograms represents the median (green, putative interneuron median = 16.67 ms; pink, putative pyramidal median = 33.33 ms). **d** Distribution of shuffle-controlled, head movement-modulated cells according to cell type (left, putative pyramidal; right, putative interneuron), cells can be upmodulated when the firing rate significantly increases, downmodulated when the firing rate significantly decreases, and down/upmodulated when they show both significant increased and decreased phases (regardless of order).

spanning 50 ms, within a 400 ms window (total of eight bins). The center of the window was aligned with the peak velocity of each detected head movement event in the recording session, with a range of 200 ms before and after the head movement peak velocity. Cells were classified as upmodulated if the real mean firing rate exceeded the 97.5th percentile of the mean shuffle firing rate at that bin and across bins, downmodulated if the mean firing rate was lower than the 2.5th percentile of the mean shuffle firing rate at any bin and across bins, and down/upmodulated if their mean firing rate both exceeded and was lower than the 97.5th and 2.5th percentile of the shuffle respectively at different time bins.

We found that a lower percentage of putative pyramidal cells (72/204, 35.3%) were modulated (Fig. 8c) in comparison to putative interneurons (105/127, 82.7%). Furthermore, the average firing rate maxima of all modulated putative interneurons peaked at 31.2 ms after the alignment 0 point (time from head movement velocity peak), whereas for all modulated putative pyramidal neurons it peaked at 81 ms, a difference of 49.8 ms. When accounting for the type of modulation (Fig. 8d), putative pyramidal cells seem more evenly split between downmodulation (14.7%) and upmodulation (19.6%), whereas, in modulated putative interneurons, upmodulation was most prevalent (66.9%), in comparison to downmodulation (6.3%). For both cell types, down/upmodulation was relatively rare at 1% in putative pyramidal and 9.5% in putative interneurons. The differences in activation latencies and the proportion of the different modulation types suggest that phase resetting may be causally linked to interneuron activation that is followed by a range of modulation in pyramidal cells[91–94].

## Discussion

We used motion capture to track the behavior of freely moving marmosets foraging in a 3D maze and recorded the responses of neurons in the CA3 and CA1 regions of the hippocampus. We found that marmosets explore the environment using alternations of body translations and frequent stops. During translations head rotations are constrained. During stops, marmosets make frequent rapid head–gaze shifts to explore the visual scene. This strategy differs from the one of rats, that frequently move their heads at low velocity "scanning" the environment as they locomote. We found that putative pyramidal neurons in the CA3 and CA1 regions mainly encoded a mix of view, head direction and place. Putative interneurons encoded a mix of AHV and TS. We demonstrated that the position of the animal in the maze (place) can be decoded from small ensembles of mixed selective cells, despite the predominance of encoding of view and head direction. Finally, we found that rapid head movements reset the phase of theta oscillations in the hippocampus. Theta phase resetting is accompanied by the activation of interneurons followed by a variety of modulations in pyramidal cells.

### Exploration strategies in marmosets and rats

Rats and mice have been the primary animal models used to study spatial navigation[95–98], with 1337 indexed publications in 2022 (Pubmed indexed search query: "Rodentia" AND "hippocampus/physiology"). Studies in NHPs have been scarce (4 in 2022 in Pubmed indexed search, query: "primates/physiology" OR "*Callithrix*/physiology" OR "*Macaca mulatta*" AND "hippocampus/physiology"). There has been a tacit assumption that results from studies in nocturnal rodents such as mice and rats can be extrapolated to diurnal primates such as macaques and humans. However, there are differences between sensory systems and behaviors between nocturnal rodents and diurnal NHPs that may be aligned with differences in how the hippocampus processes information during navigation. For example, rats and mice lack foveae, their eyes are displaced to the sides of the face, and they lack the wide range of color vision of diurnal primates like marmosets[31,99]. Rats can move their eyes independently, while marmosets cannot; marmosets make conjugate eye movements that preserve the alignment of the two eyes needed for stereovision[31,49].

Marmosets' high acuity color and stereovision allow far sensing during daylight to forage for fruit, trees, discriminate between conspecifics during mating or social interactions, or escape predators. These specializations make them distinct from nocturnal rodents such as rats.

Rapid head movements are key to marmosets' and other diurnal primates' efficient exploration of the environment via the far-sensing capabilities of their visual system[100]. They have previously been described in marmosets sitting in a primate chair[44]. In this study, we report the presence of these highly stereotypical movements in a marmoset navigating in 3D space. They followed the main sequence, much like it has been described for eye–head gaze shifts in macaques[51] and predominantly occur when the subjects are stationary, visually exploring the environment (visual navigation). We reasoned that because vision is "far sensing" and the high-resolution fovea allows for the estimation of objects' features, animals do not need to actively visit locations to identify potential targets and landmarks for navigating. They can also identify depth stationary cues such as size differences, objects' occlusion cues, and dynamic cues using motion parallax that allow evaluating distances to target and path planning before translating in 3D space[101,102].

In contrast, mice and rats use a different strategy likely adapted to their nocturnal lifestyle. Their limited color vision, lack of a fovea, and the absence of daylight may have evolved in the use of near-sensing capabilities to identify target objects during navigation and the lack of rapid head–gaze movements. Instead, they use slower head movements during navigation likely to position the whiskers or orient their olfactory apparatus toward possible targets or landmarks. In the presence of poor illumination, rats would more frequently need to use path integration during self-motion for exploring the environment relative to marmosets.

The divergence between primates and rats may be traced to their ancestors. During early evolution, when mammals were predominantly nocturnal to escape predation by dinosaurs, most mammals regressed their visual capabilities, more suited for daylight, while expanding somatosensory, olfactory, and hearing capabilities better suited for nocturnal activities[100]. However, primates may have escaped the pre-extinction nocturnal bottleneck by developing a sophisticated visual system, expanding the pathway from the retina to the thalamus and the visual cortex. This reliance on high-resolution stereo vision allowed foraging for insects and fruit in the distal branches of trees but may have produced a regression of their olfactory and somatosensory capabilities (e.g., whiskers). After the extinction of the dinosaurs, 66 million years ago, primates safely invaded the day-life niche further and disproportionally developed their already expanded visual system to incorporate color vision and extraordinary stereo and object recognition abilities[100]. This may have led primates, such as marmosets, to adopt more efficient navigation strategies relying on the far-sensing power of vision and consequently shaping neuronal selectivities in the hippocampus. In contrast, rats never escaped the nocturnal bottleneck and therefore, like their mammalian ancestor, preserved their near-sensing capabilities such as whiskers, olfaction and audition to support navigation. Indeed, our data indicate that the common marmoset, a diurnal primate, uses different strategies to explore and navigate the environment compared to the rat, a nocturnal dweller. These differences may have impacted the physiological mechanisms of spatial navigation and specializations in the hippocampus of the two species. Further experimentation using similar behavioral paradigms and readouts (habituation times, mazes, tasks) between both species would be beneficial to systematically bridge this gap.

### Representation of space in the marmoset hippocampus

Previous studies in rats have shown that place cells form the basis for an internal representation of a spatial cognitive map[1,4]. The percentage of all hippocampal pyramidal cells that can be considered place cells ranges from 20 to 25%[4,103]. The identification of place cells in the rat hippocampus is astonishingly clear and replicable[1,4]. Studies in other species such as nocturnal bats have also reported place cells[87,104]. In primates, reports of place cells are scarce. Instead, studies using virtual reality in macaques have shown a representation of spatial locations that depend on objects-context[32] or landmarks[105]. A recent study in freely moving macaques[16] found that classic place cells are rare (7% of all cells). They reported that about 26% of all cells encoded place, but other variables such as facing location were better encoded. The only study in marmosets[17] showed that when subjects moved on a linear "L-shaped" track, 14.1% of cells could be classified as place cells. Interestingly, they reported that 77.9% of these cells were directionally selective, which could be interpreted as encoding head direction or view, as in the macaque study[16].

Our study was conducted in freely moving marmosets in a 3D environment, where these small primates naturally forage. Although we found neurons that encoded view and place in the 3D maze, we show a predominance of variables related to the head direction and view. Place encoding occurred in 30.4% of all cells; however, we observed that place was exclusively encoded in conjunction with either view or head direction, and most frequently both.

Our results may be related to changes in the brain structure and function happening in diurnal primates such as the expansion of areas related to vision, and the emergence of a high-resolution fovea[100]. Diurnal primates have also evolved a sophisticated eye–head apparatus that allows coordinated gaze shifts to stabilize the fovea on objects of interest. Vision, as a far-sensing strategy has shaped diurnal primates' lifestyle and the physiology of the hippocampus. From an anatomical connectivity perspective, the primate hippocampus receives more visual information than that of the rat[106–110]; from a behavioral perspective, as shown in this study, there is a prevalence of navigation strategies that favor visual exploration behavior in primates. We argue that primate brains developed navigation and spatial memory systems adjusted to their diurnal lifestyles that heavily rely on visual cues and landmarks rather than on maps of space. One issue that remains unclear is whether the entire range of selectivities described as part of the navigational GPS in the entorhinal cortex of rodents (e.g., grid cells, border cells) is also present in primates. One study in macaques has described grid cell-like gaze selectivity in the entorhinal cortex when subjects inspected a visual scene[111]. This, however, is very different from grid cells that triangulate the animal's position in the environment.

We found no evidence of cells that encoded place independent of view or head direction. As described above, marmosets exhibit a preference for visual exploration over physical visits to locations, this can potentially introduce biases in the representation of spatially relevant variables. These biases might account for some of the differences in spatial encoding we observe with respect to rodents. We recognize the necessity for different experimental controls to ensure a more equal sampling of space and view.

It is important to note that in our study, occupancy was biased toward reward sites, which are at fixed locations in the maze. This could have had implications in how place and view was represented in neuronal populations, and it is different from conventional definitions of place cells in rodent literature. In classical rodent experiments, investigators randomly scatter rewards throughout the maze, and spatial selectivity analyses are limited to translation movement epochs (SIC analysis with only translation movement epochs can be found in Fig. S4a). In our experiments, we did corroborate that each view field from all view selective cells was sampled from at least three different locations (Fig. S4c). In fact, we found that most view fields were sampled from 10+ unique locations that were spread apart from each other as far as 100 cm (Fig. S4d) and were viewed from different angles (60°+, Fig. S4e). Exploration biases seem to be a challenge when testing animals on volumetric mazes, particularly with regard to spatial sampling along the vertical dimension, as non-flying animals tend to

use different strategies to travel horizontally than vertically[15,112,113]. Additional experimental controls might be needed to better disentangle the contribution of 3D volumetric place and view on neuronal firing, and its contrast with occupancy.

Additionally, the median SIC corresponding to view is significantly higher during epochs of slow or no head movement when compared to epochs of rapid head movement (two-sided Wilcoxon signed-rank test, $p = 8.29e-15$, $Z = 7.76$, Fig. S4b). This suggests neurons may respond to visual targets being observed during fixations rather than the mere act of shifting gaze through space. It also raises questions as to whether this response stems from memory, object recognition, or purely spatial view encoding. Furthermore, view projection on the maze's walls does not account for possible encoding of local landmarks and cues situated inside the maze (climbing ropes, climbing platform and access holes, see Supplementary Movie 1, Fig. S2a) that may be the target of fixation. We did as much as possible to minimize those landmarks but further investigation is needed to control for these factors and comprehensively determine local landmark contributions to neuronal firing.

### Representation of speed in the marmoset hippocampus

While there is abundant evidence of speed-correlated activity in the hippocampus of rodents[11,12,58,68–73], there is limited evidence in primates. A study[16] reported significant encoding of both linear speed and angular velocity (TS and AHV in our study, respectively) in freely moving macaques, where most of these speed-encoding cells were putative interneurons. Similarly, a study found hippocampal cells in macaque monkeys, that responded to linear and rotation-assisted motion (monkey sitting in a remote-controlled robotic platform)[114], suggesting there are both vestibular and optic flow inputs to these speed-encoding cells. Our study reports speed cells in the freely moving marmoset in a 3D environment. Moreover, we show that AHV was encoded to the same level as TS and that most cells had mixed selectivity for both variables. In agreement with rodent literature, the strongest speed-correlated signal was observed in putative interneurons[58].

One notable difference from findings made in rodents is the prevalence of AHV encoding in marmoset CA3/CA1 neurons. Encoding of AHV has been reported in rodent studies almost exclusively in either parahippocampal regions (MEC, parasubiculum, presubiculum)[115], retrosplenial cortex[116], or sub-cortical regions associated with vestibular information processing (lateral mammillary nuclei, thalamic nuclei and striatum)[7,8,117,118]. The encoding of AHV has been theorized to serve an essential role in the generation of grid cells[119,120] or the general processing of self-motion[74,121,122]. The function of head-speed cells in the marmoset is not clear but may be related to the signal that produces head–gaze theta phase resetting.

### Ensemble coding of space in the marmoset hippocampus

Our results demonstrate that space can be decoded from a pseudo-population of neuronal ensembles' firing activity in the hippocampus regions CA1/CA3 of marmosets (Fig. 7). Remarkably, the neurons that yield the highest decoding accuracy, are not highly selective to specific places. Instead, these neurons exhibit mixed selectivity for view, head direction and place (Fig. 7b, e). Numerous prior works have explored how neuronal mixed selectivity supports efficient representation and processing of complex information in the brain[76,77,123]. Mixed selective neurons exhibit modest selectivity for individual features, but increased selectivity for combinations of two or more features[123]. It has been theorized that neuronal ensembles face limitations to flexibly represent feature dimensions across different behavioral contexts when neuronal selectivity is highly specific[123]. This limitation becomes particularly prominent when considering that ensembles are constrained by a finite number of neurons. To overcome these limitations, neuronal networks may leverage between specificity and flexibility of neuronal selectivity[77]. This approach is especially advantageous when high-dimensional representations are required. Coding of space can be regarded as highly dimensional when encompassing a multitude of sensory inputs[66,124,125]. We propose that the presence of highly mixed information in the marmoset hippocampal neuronal ensembles supports reliable representations of space.

One contrasting difference between the neuronal responses observed in this study (and the NHP hippocampus in general), and the ones commonly observed in nocturnal rodents, is the predominance of variables directly related to gaze such as view and head direction[16,18,20,35]. Marmosets high-resolution foveal vision might allow anchoring of place representations and future paths for navigation to visual/scene cues, such as landmarks identity, depth, egocentric/allocentric location and the spatial relationships between them extracted from visual exploration via gaze shifts[21]. In contrast, "near" sensory cues like olfaction and whisking are readily available to nocturnal rodents such as rats. With heads positioned much lower above the ground relative to marmosets and a nocturnal lifestyle prioritizing places with poor illumination, rats may be "less" visually driven. However, recent work has demonstrated diverse selectivity in the hippocampus of rats; implementation of multivariate encoding analyses and novel experimental paradigms in previous studies provide evidence that place cells in the hippocampus of rats also encode variables beyond place, like position, distance and direction of motion of a bar of light under body fixed conditions[126], or be modulated by head direction, the presence of visual and olfactory cues or immediate experience[12,125,127–129]. Evidence from multiple species has consistently found a high degree of mixed selectivity in hippocampal neurons, observed across multiple species: rodents[127,128,130–132], bats[133], rhesus macaques[16], marmosets[134] and humans navigating in virtual reality[135]. Thus, mixed selectivity may be the norm in hippocampus neurons. However, mixed selectivity seems to be biased to overrepresent certain variables depending on the ecological niche of the species. For the case of marmosets, encoding of visual variables related to gaze orientation seems to be predominant in hippocampal neurons, at least in the daylight conditions in which diurnal primates usually forage.

Indeed, we found that place can be decoded from ensembles with a predominance of view and head direction mixed selective neurons. However, the decoding accuracy is lower than the one reported in rodent literature[78,136], where the decoded spatial resolution is often lower than 5 cm. It is possible that our analyses might have been limited by the size of the decoded spatial bin (30 cm × 60 cm), which is due to the need to include sufficient samples of space. The information provided by these ensembles about landmarks in the environment (allocentric coding) and self-orientation relative to them (egocentric) can be sufficient to position oneself in space (origin of the place information) and aid goal-directed navigation[137] and the formation of spatial memories. It may also provide flexibility to adjust navigation strategies to the task demands (e.g., egocentric rather than allocentric representations[32]).

### Head–gaze movement phase resetting

It has been shown that memory encoding involves interactions between theta oscillations and incoming sensory signals into the hippocampus[138]. Indeed, responses of hippocampus neurons to sensory events are synchronized to a certain phase of theta oscillations[139,140]. It has been proposed that theta oscillations are like a metronome for coordinating sensory information transfer from cortical areas to the hippocampus[138]. This theory matches data from rodents; however, it has a shortcoming when extrapolated to humans. In humans and NHPs, hippocampus theta oscillations are not rhythmic but appear in short bouts[16,17,141,142]. Interestingly, some studies have reported that theta oscillations are locked to the execution of saccades[42], a phenomenon known as saccade phase resetting[40].

Rapid head movements in species of primates are used to direct gaze toward spatial locations[29,51]. In marmosets head velocities reach values above that of eye velocities during saccades made with the head restrained (Fig. 2c). Here we report an LFP modulation described as head–gaze theta phase resetting[27,42,89,143,144]. Moreover, a significant percentage of neurons' firing rate was modulated during and following head movement initiation. Notably, in putative interneurons the most common form of modulation was an increase in firing rate (upmodulation), but in putative pyramidal cells, it was evenly split between up and downmodulation (Fig. 8d). The peak of the modulation effect was also found to be different across cell types, where putative interneuron modulation peaked before putative pyramidal cell (49.8 ms faster, Fig. 8c). Together, this suggests that the signal triggering head movements (e.g., a corollary discharge (CD) signal) initially activates interneurons that may reset background noise in pyramidal neurons that can be differentially activated by incoming sensory inputs. In our analyses, we aligned neural data to head peak velocity because we did not have estimates of eye movement latency. It is well known that saccadic eye movements modulate LFP power and phase, as well as neuronal firing in the primate hippocampus[40,89,91]. We observed modulations of both LFP and firing rate -120 ms before the head peak velocity (Fig. 8a–c) which could correspond to a similar phenomenon to the one observed during saccades, which classically exhibit shorter latency compared to head movements[51]. Future experiments combining eye tracking with freely moving head recordings might help elucidate the latency profiles of the modulation found in our work and its relationship to saccades and head movements.

We propose that head–gaze theta phase resetting may act as a "single pulse metronome" that synchronizes stochastic firing in neuronal ensembles before the eyes land on a target, allowing incoming sensory signals to be "distinguished" from the background noise of the circuit. The activation of inhibitory interneurons may be causally linked to the process of resetting neuronal ensembles and decreasing noise, while the modulation in putative pyramidal neurons may be directly related to the arrival of sensory inputs to ensembles of pyramidal cells and encoding of sensory signals. This mechanism may be the driver of theta oscillations in diurnal mammals with foveal vision that explore the environment through voluntary gaze shifts. It has been further proposed that during a gaze shift a CD signal originating in the SC reaches the thalamus and then via the nucleus reuniens reaches the hippocampus[91,94]. However, a CD signal that reaches areas of the neocortex seems to do so via the medial dorsal thalamic nucleus[145] and plays roles such as inhibiting visual processing during saccades and remapping of receptive fields[145–148]. It is currently unclear how the CD signal in the hippocampus and neocortical areas are linked and how it relates to the head–gaze phase resetting phenomenon reported here.

Our results provide evidence that freely moving marmosets use different exploration–navigation strategies compared to rats. These strategies have shaped physiological adaptations in the hippocampus. Cognitive maps of space in the marmoset and likely in other diurnal primates may be driven by mixed or conjunctive coding of gaze-related variables that enable encoding of visual features and object relationships used as landmarks for navigation. Head–gaze phase resetting seems to play a role in synchronizing theta oscillations to increase the efficiency of information encoding in the marmoset hippocampus.

# Methods
## Statistics and reproducibility
A description of statistical techniques and a table of tests used in the main figures has been provided in the supplement. No statistical method was used to predetermine sample size.

## Ethics statement
The animal care and handling procedures, encompassing basic care, housing and husbandry, animal training, surgical procedures and experiments (data collection), were granted approval by the University of Western Ontario Animal Care Committee. This approval guarantees adherence to federal (Canadian Council on Animal Care), provincial (Ontario Animals in Research Act), regulatory (e.g., CIHR/NSERC) and other national CALAM standards, ensuring the ethical use of animals. The animals' physical and psychological well-being was regularly assessed by researchers, registered veterinary technicians, and veterinarians.

## Animals
Two adult (1 female, aged 5 y and 1 male, aged 3 y) marmosets (*Callithrix jacchus*) were utilized for all experimental procedures described in this study. The subjects were paired-housed in custom primate cages located within the primate facility at the Robarts Research Institute. A 12-h light cycle was maintained (Day: 7 a.m.–7 p.m., Night: 7 p.m.–7 a.m.).

The subjects were provided with a standard diet of dry food formula supplemented with fruit, nuts, and various protein sources. To facilitate positive interactions and ease of handling, subjects were gradually acclimated to gentle glove handling by the experimenters. Additionally, they were trained to enter plexiglass transfer boxes and navigate through the experimental setup using positive reinforcement, obtaining a liquid reward of condensed milk or gum arabic (*Acacia*), administered either directly by the experimenter or delivered via metal cannulas placed inside the maze.

## Experimental setup and behavioral paradigm
Subjects were trained to forage for reward in a rectangular transparent maze with three different vertical levels (3D maze, Fig. 1a) for at least 2 weeks before recording. For the majority of the recording sessions, the subjects were placed in the chamber alongside their cage partner with the purpose to alleviate stress. Reward locations were placed at four different positions in each level of the maze, two on one side and two on the opposite side, resulting in a total of 12 reward locations across the maze (Fig. 3b). Each reward location consisted of a small panel containing an LED and a liquid reward delivery system controlled by a solenoid via the NIMH MonkeyLogic[149] software. The subject was cued by flashing the LED. When the subject approached the LED location and was closer than 15 cm to the reward delivery site a sound was played, and a reward was delivered. After that another, different LED was flashed, and the sequence of events was repeated. We also rewarded the subjects manually by waving a hand at the reward location and delivering a marshmallow when the subject approached. Recording sessions lasted for as long as the subject was willing to continue foraging (-40–60 min).

## Motion capture 3D tracking
A total of 14, synchronized, cameras (Optitrack, Flex 13, Corvallis, OR), were placed around the 3D maze in a configuration that optimized coverage and minimized occlusion. Video frames were acquired at 60 Hz. The cameras emit 850 nm infrared (IR) light and are fitted with a lens filter for IR light.

A recording chamber was placed on the skull of the subjects that allowed to house the electrodes and wireless recording headstage, a 3D printed cap covered the recording equipment and six spherical retro-reflective markers (4 mm diameter, Facial Marker, Optitrack) were arranged in a unique geometric configuration (rigid body) and affixed on top of the cap.

Before every recording session, the setup was calibrated using a calibration wand (CWM-125, Optitrack). The wand consists of three markers of known dimensions and distance to each other that

are used to compute a volumetric world coordinate system measured in metric units via triangulation algorithms. A calibration square (CS-100, Optitrack) was used to define the zero point (where the horizontal or $X$, depth or $Y$, and vertical or $Z$ axes intercept) and the plane corresponding to the ground (plane parallel to the maze floors); the calibration square and subsequently the zero point was always placed at the lowest point of the south–west corner of the maze. This calibration will be used by the tracking software (Motive version 2.0.2, Optitrack) to compute the position and orientation of the cameras; the calibration step is essential to triangulate the 3D position ($X_p$, $Y_p$ and $Z_p$, in metric units) of the markers from the individual camera's 2D images.

After the 3D position of the cluster of cap markers is estimated, the rigid body is tracked as a unique object at the pivot point (origin of the rotational axes roll, pitch and yaw), this point is manually placed at roughly the intersection between the vertical axis of the neck of the subject and the visual axis (eye level). The roll axis is manually aligned so that it is parallel to the estimated visual axis (red axis, Fig. S1a, b), the pitch axis is manually aligned parallel to the maze floors (estimated using the ground plane, blue axis, Fig. S1a, b) and the yaw axis is aligned parallel to the vertical plane (plane perpendicular to the ground plane, green axis, Fig. S1a, b).

## Implantation surgery and electrophysiological recordings

Prior to every surgical or imaging procedure, subjects had their food removed at a maximum of 4–6 h before the procedure while maintaining free access to water. For short-duration procedures like CT imaging, subjects were sedated via intramuscular injection of Ketamine (15 mg/kg) and Medetomidine (0.0125 mg/kg). For longer-duration procedures, like MRI imaging and implantation of recording chamber and electrodes, we used the same sedation protocol described above for induction of anesthesia. Maintenance of anesthesia was then achieved through continuous intravenous administration of Propofol (0.3 mg/kg/min) or via inhalation of Isoflurane (0.5%–2.0%). In certain cases, a combination of both maintenance drugs was necessary. Post-operative analgesia was administered for a minimum of three days using intramuscular Buprenorphine (0.01–0.03 mg/kg) once to thrice daily and oral Acetaminophen (6–10 mg/kg) twice to four times daily, analgesia was continued beyond the initial three days as deemed necessary with Acetaminophen only.

After co-registration of anatomical MRI (9.4T) and micro-CT (150 μm) scans[150–152] (3D Slicer software version 4.10, slicer.org), we implanted a recording chamber "cap" (Fig. S1i, j) that allowed a 3D printed grid to be placed above the skull surface, each space in the grid could be used as a fiducial to calculate the trajectory necessary to reach the CA fields. We chronically implanted 32ch microwires "brush" array electrodes in the hippocampus' CA1 (Subject C) and CA3 (Subject P) (Microprobes, Gaithersburg, USA). The final position of the electrodes was verified with post-surgical micro-CT imaging and registration to the pre-surgical anatomical MRI (Fig. 3a; Fig. S2b). In Subject P, we used a semi-chronic electrode implant that consisted of a small microdrive that allowed the electrode to be lowered after implantation (2 mm total range). We lowered the microdrive in increments of 70–150 μm, until we reached the end of the working range and finally, took micro-CT images to verify the final position of the electrode.

Neural activity was recorded using wireless telemetry (CerePlex Exilis, Cerebus Data Acquisition System, Blackrock Neurotech, UT, USA) sampled at 30 kHz.

## Data analyses and visualization

All analyses were performed using custom-built MATLAB scripts (Mathworks Inc, version 2021b) unless otherwise specified. A MATLAB wrapper was used to calculate the GAMs using PyGAM[75], a Python open-source resource. Colormap BrewerMap was used to visualize LFP power spectrum[153,154].

## Body translation speed

TS was defined as:

$$s = \frac{\sqrt{(\Delta x)^2 + (\Delta y)^2 + (\Delta z)^2}}{\Delta t} \qquad (1)$$

where $s$ is the TS of the rigid body, $x$ is the position in the horizontal axis, $y$ is the position in the depth axis, $z$ is the position in the vertical axis and $t$ is time.

## Angular head velocity

The rotation vectors (pitch, raw and roll) were exported from the motive software as rotation quaternions ($q$) of unit length, defined as:

$$q = q_0 + iq_1 + jq_2 + kq_3 \qquad (2)$$

where $q_0$, $q_1$, $q_2$ and $q_3$ are real numbers, and $i$, $j$ and $k$ are imaginary unit vectors. Where the real part is a scalar and the imaginary parts the vector. In a system with three rotational axes (such as pitch, yaw and roll), the alignment of two axes restricts the third. Where rotations along the locked axes produce the same effect as rotations along the combined axis, this is commonly referred to as gimbal lock. Quaternion constructs avoid gimbal lock by representing space as a three-dimensional sphere in four-dimensional space where any loop along its topology can be continuously contracted to a single point[155], that is, any two unit quaternions representing rotations can be connected by a continuous path within this space and are considered simply connected. Quaternions offer an advantage over traditional Euler's axis angles in that they are a more efficient way to compute rotations and are not susceptible to gimbal lock.

The AHV was defined as:

$$\omega = \frac{\triangle \theta}{\triangle t} \qquad (3)$$

where $\omega$ is the AHV, $\theta$ is the angular distance and $t$ is time.

The angular distance $\theta$ between $\theta_{t2} - \theta_{t1}$ was computed using the MATLAB function *dist* (Robotics and autonomous systems toolbox).

## Significant head and body movement classification

Significant head movements were defined as epochs where AHV was higher than 200°/s and the amplitude of the movement was higher than 10°. Movements with velocities higher than 2000°/s were deemed artifactual.

Significant body translations were defined as epochs where the TS was higher than 16 cm/s and the amplitude of the movement was at least 30 cm. Movements with speeds higher than 300 cm/s were deemed artifactual.

## Naka–Rushton function fit

We used MATLAB's curve fitting function *fit* (non-linear least squares implementation), to estimate the parameters $R_{max}$, $n$, $K$ and $b$ of a Naka–Rushton function:

$$R(X) = R\max\left(\frac{X^n}{X^n + K^n}\right) + b \qquad (4)$$

where $X$ is the predictor variable, $R(X)$ is the response, $R_{max}$ is the peak response, $n$ corresponds to the exponent, $K$ is the number of $X$ at which $R(X)$ reaches half of its maximum and $b$ is an additive constant.

## Head Direction

We considered the distinction between head direction and view as illustrated by ref. 156 (Fig. S1h).

The head direction is solved by the tracking software using a custom arrangement of markers that creates a rigid body with pre-defined *XYZ* axis angles. The axes are aligned to the subject's head anatomy and a pivot point is placed virtually in the neck, in a way that the horizontal head direction of the rigid body runs along the midline of the head and is parallel with the head direction and the vertical axis perpendicular to the maze floors.

The tracking software solves the head direction in terms of angular values of yaw (horizontal axis), pitch (vertical axis) and roll (lateral tilt axis) for the rigid body as a whole, in an allocentric reference frame.

The 2D head direction is estimated from the yaw angle values alone, and the 3D head direction is estimated from yaw, pitch and roll angles.

### View as facing location

A ray is cast from the vector of 3D head direction and the intersection of this ray with the walls of the maze is defined as the facing location. Since the marmoset's oculomotor range is largely <10°, we interpret the facing location as view[31].

### Signal pre-processing and spike sorting

Offline spike sorting was performed using Plexon (Offline Sorter version 4.5.0, Plexon Inc., TX, USA), a 4-pole, Butterworth, 250 Hz high pass, a digital filter was applied to the raw 30 kHz broadband signal, and a −4 sigma noise threshold was used to detect and align individual waveforms. Principal component analysis was used to define the feature space and automatic (T-Distribution E-M) sorting was applied to isolate units. Manual inspection was then performed to classify units into noise units, multi-units or single-units (Fig. S2c–e). Quality control measures were implemented after sorting the spike data. Single units that fired <100 spikes per session were invalidated. Additionally, to account for the use of microwire arrays where the final location of individual wires cannot be controlled, potential duplicate units were invalidated if the isolated units shared more than 50% of the spike times (estimated at 1 ms temporal resolution). In cases where two or more units met this criterion, only the unit with the highest signal-to-noise ratio was included in the subsequent analyses.

### Burst Index

Bursts can be described as a rapid sequence of action potentials followed by a period of relative quiescence[157,158], it has been estimated that 95% of pyramidal cells in the macaque monkey CA3 fire in bursts[159,160]. The burst index (BI) is a metric that has been proposed to describe the propensity of burstiness in the firing pattern of neurons, where higher BI values indicate a neuron's higher propensity to fire in bursts. As previously described[60,161] we calculated the BI by estimating both the inter-spike interval (ISI) histogram and the predicted ISI distribution, which is based on a Poisson distribution calculated from the mean firing rate of the whole recording session.

The predicted Poisson distribution of ISI was computed as follows:

$$f(t) = \lambda e^{-\lambda t} \qquad (5)$$

where $\lambda$ = firing rate and $t$ = time bin. The probability was calculated for each 1 ms time bin in the 2–40 ms time range, a normalization procedure was implemented by summing all of the predictions in both the predicted and real ISIs.

The BI was computed as the following:

$$\text{burst index} = \frac{\sum \text{ISIs}_{\text{measured}} - \sum \text{ISIs}_{\text{predicted}}}{\sum \text{ISIs}_{\text{measured}} + \sum \text{ISIs}_{\text{predicted}}} \qquad (6)$$

where the sum of ISIs was performed between the values from 2 to 20 ms. The BI is ultimately defined as the division between the subtraction of the measured ISI sum and the ISI predicted, and the sum of both these sums. In the end, the index is bound between −1 and 1.

### Classification of putative interneuron and pyramidal cells

Macaque monkey hippocampal neurons have been putatively classified as interneuron/pyramidal cells based on firing rate alone[61,63], where interneuron cells are typically of the fast-spiking type (average firing rates above 10 Hz). To increase the robustness of classification in our analysis and avoid potential species-specific biases that firing rate thresholding classification could introduce (since it has only been used in macaque monkeys), we implemented an unsupervised classification algorithm (*k*-means, MATLAB function *k*-means) using BI and average firing rate as features. We refrained from using other classical methods of classification like spike waveform width since it has been shown that short-duration waveforms from neurons recorded from the hippocampus can represent axonal activity, and are not necessarily inhibitory interneurons in nature[62].

### Spatial information content

Occupancy and firing rate maps were estimated based on a spatial bin division shown in Fig. 4b, c, (place bins $n = 84$, view fields $n = 106$) per recording session[162]. Bins that weren't sampled more than 3 independent times, or <200 ms were discarded. SIC[66,67] was calculated independently for place and view for every individual cell. SIC was defined as:

$$I = \sum_i^L P_i \frac{\lambda_i}{\bar{\lambda}} \log_2 \frac{\lambda_i}{\bar{\lambda}} \qquad (7)$$

where $L$ is the number of bins, $P_i$ is the proportion of occupied time in each bin, $\lambda_i$ is the mean firing rate for the $i$th bin and $\bar{\lambda}$ is the mean firing rate of that cell across all bins as defined by:

$$\bar{\lambda} = \sum_i^L P_i \lambda_i \qquad (8)$$

Significant cells were defined as SIC > 0.05 (corrected alpha value for the number of bins) of the null distribution (5000 circular shift permutations). In the same way, for significant cells, significant bins (place or view fields) were defined as SIC > 0.05 (corrected alpha value for the number of bins) of the null distribution for that bin.

### Speed cell definition

A single cell was defined as a speed cell if it met two criteria:

**Speed score higher than 0.3.** The speed score is defined as the Pearson's correlation coefficient between the time series of speed and the time series of firing rate[58,74]. After determining the instantaneous speed (sampled at 60 Hz), both the time series of speed (AHV and TS were computed independently) and firing rate were smoothed using a 1-D Gaussian with 250 ms standard deviation. The correlation coefficient was defined as the following (calculated using the MATLAB function *corrcoef*):

$$r = \frac{\sum (x_i - \bar{x})(y_i - \bar{y})}{\sqrt{\sum (x_i - \bar{x})^2 \sum (y_i - \bar{y})^2}} \qquad (9)$$

where $x_i$ and $y_i$ are individual samples from the time series (e.g., $x$ can be speed and $y$ firing rate) and $\bar{x}$ and $\bar{y}$ are the mean across all samples (e.g., $x$ can be mean speed and $y$ mean firing rate). In order to allow comparisons with prior work characterizing speed-correlated firing in the hippocampus of rats[58], we used their same definitions. In that study, the threshold for categorizing a cell as a speed cell was a speed score >0.3, as it effectively separated a small distribution of highly

speed-correlated cells within the distribution of the population's speed scores, we observed a similar pattern in our dataset, where this threshold segregated a similar subset of highly speed-correlated cells for both AHV and TS (Fig. 5b).

**Shuffle control criteria.** Using the same smoothed firing rate time series, a shuffle distribution of speed scores was computed after circularly shifting the time series 1000 times. This distribution served as the null distribution used to estimate the significance of the real speed score (real speed score higher than 95th percentile null distribution).

Since AHV and TS speed scores are estimated independently, cells can be classified as AHV cells, TS cells or both.

### Generalized additive model (pyGAM)

GAMs are a particularly useful tool to explore the relationship between neural activity and multiple behavioral variables. They are specifically advantageous when exploring complex and non-linear interactions between the covariates and the independent factor[163]. Here we describe the implementation used in every neuron reported in the paper, we used this model to classify single cells according to their encoding profiles; where a cell could significantly encode none, one or a combination of behaviors (predictors).

The general form of GAMs is described by:

$$F(x) = y = \exp(\beta_0 + f_1(x_1) + f_2(x_2) + \ldots + f_r(x_r)) \qquad (10)$$

In our implementation $y$ is a single neuron's firing rate, $\beta_0$ is the intercept (constant), $[X_1, X_2, \ldots X_r]$ are the predictor variables (space, view, head direction and speed).

This GAM implementation has three components: distribution, link function ($F(x) = g(\mu|X)$) and a functional form ($f_r$). We calculated all our models using a log link function, Poisson distribution and a functional form function of penalized B splines[164]. The models are cross-validated during training to avoid overfitting, and the goodness of fit estimates are averaged across $k$-folds.

As reported in the results section, spatially selective cells are mainly putative pyramidal cells and speed-encoding cells are mainly putative interneurons (Fig. 5d), so in order to optimize computing times, we designed two separate models, one for putative pyramidal neurons (including space, view and head direction) and the other for putative interneurons (AHV and TS).

We utilized a nested, stepwise, forward search model selection approach to determine the optimal number of predictors per cell (Fig. S5a). The methodology involved a gradual construction of the model by iteratively incorporating predictor variables based on their individual contributions to the overall goodness of fit. Initially, all available one-variable models (first-order models) were fitted, and a surrogate null distribution of first-order models was computed. This null distribution was obtained by randomly circularly shifting the vectorized firing rate a total of 100 times. To establish statistical significance, we compared the actual explained deviance (ED) with the 95th percentile of the null distribution (Bonferroni corrected). ED is described as the following ratio:

$$ED = 1 - \left(\frac{D_{null} - D_{fitted}}{D_{null}}\right) \qquad (11)$$

where $D_{fitted}$ is defined as:

$$2\left(\sum_{i=1}^{n} y_i \log\left(\frac{y_i}{\hat{\mu}_i}\right) - (y_i - \hat{\mu}_i)\right) \qquad (12)$$

where $n$ is the number of observations, $y_i$ is the actual spike count for the $i_{th}$ observation, and $\hat{\mu}_i$ the predicted mean spike count for the $i_{th}$ observation.

$D_{null}$ is defined as:

$$D_{null} = -2(\text{LogLikelihood}_{saturated} - \text{LogLikelihood}_{null}) \qquad (13)$$

The log Likelihood$_{saturated}$ is defined as:

$$\sum_{i=1}^{n} y_i \log(y_i) - y_i - \log(y_i!)) \qquad (14)$$

and the Log Likelihood$_{null}$ is defined as:

$$\sum_{i=1}^{n} y_i \log(\hat{y}) - \hat{y} - \log(y_i!)) \qquad (15)$$

where $\hat{y}$ corresponds to the mean firing rate (constant).

If the real ED exceeded this threshold, the model was considered statistically significant.

The ED of the real model was then normalized by subtracting the mean shuffle ED from the real ED, allowing us to select the first-order model with the highest ED when multiple first-order models crossed the significance threshold. The variable in the best model was retained as a predictor for the subsequent iteration of model selection ($n + 1$ order model). In each iteration, one of the remaining variables was added at a time until all combinations of that order were evaluated. Similar to the first-order models, statistical significance was assessed by comparing it against a null shuffle distribution, which is calculated at every iteration of the training process, via randomly shifting the vectorized new added variable. For models with more than one variable, the time series of the single predictor under consideration was circularly shuffled, while the firing rate and already selected predictors remained unaltered. The training process concluded when the incremental improvement in model fit by adding more variables became statistically negligible. At this point, the final combination of variables was selected as the best model for a specific neuron.

The performance of the winning model is evaluated by generating a spike raster prediction using the best final nested GAM model (Fig. S5b, c). These predictions are then compared to the actual neuron's rasters using stratified cross-validation with fivefolds. Since each recording session (40–60 min long) is divided into five equally timed folds, each predicted raster is a continuous time series ~8–12 min long. To assess the similarity between the predictions and the real rasters, the Pearson correlation coefficient is calculated between the predicted and real raster data. Additionally, the coefficient of determination ($R^2$) is obtained for each fold, and the final $R^2$ is computed by averaging across all folds (Fig. S7a–c).

To determine the statistical significance of the model's predictive power, an $F$-test is performed on the calculated $R^2$ values, using an $\alpha$-value of 0.05. This test allows us to ascertain if the model's performance is significantly better than what would be expected by chance. Cells whose winner model's performance is statistically significant are considered to encode the variables used as predictors in that model.

### Place decoding analyses

We used a linear multi-class SVM classifier (*fitecoc*, MATLAB function), composed of "$n$" number of classes of binary learners in a one-vs-one design. The performance accuracy of each model was evaluated using fivefold cross-validation and trained ten times using distinct subsamples of all available trials. The subsampling procedure ensured the proper balance of classes with varying trial numbers. The model's features were derived from firing rates of a pseudo-population of neurons, obtained from periods where the AHV was lower than 200°/s, akin to head "fixations", which in marmosets can be considered analogous to eye fixations. The firing rates were integrated over 200 ms centered around those low AHV periods; each interval was treated as a trial. We included neurons in the pseudo-population if they had samples from a spatial bin (see below) at least 50 times (trials) per recording session.

To decode place from this pseudo-population, we partitioned the maze into two spatial bins per floor, resulting in a total of six spatial bins. We identified the four spatial bins (the model's classes) with the highest visitation frequency and number of neurons that could be incorporated into the pseudo-population (Fig. 7a, d).

We used a previously described ensemble construction procedure[81,82], this procedure iteratively finds the combination of neurons that provides the best decoding performance from a pool of pseudo-population of neurons (Fig. S7e). First, we train all the single-neuron models and identify the model with the highest decoding accuracy. We use this neuron as the "seed", and we fit all possible combinations of $n + 1$ ensembles including the seed and the remaining neurons in the pool. We then select the sub-ensemble with the highest decoding accuracy and continue fitting $n + 1$ models with the remaining units. We repeat this procedure until the performance of the model becomes asymptotic and does not significantly improve by adding more units; this final ensemble was referred to as the "all units best ensemble". To evaluate the contribution of the neurons that significantly encode spatial variables, as per the GAM analysis, we repeated the previous ensemble construction procedure while excluding those neurons from the pool. We referred to this ensemble as the "no significant units best ensemble". Finally, we fit a Naka–Rushton function to both ensembles' performance, to calculate statistics on the fit coefficients and to aid with visualization. Raw ensemble performances can be found in the Supplementary Information (Fig. S8a, b).

To validate the efficacy of the ensemble construction procedure in identifying the "optimal" combination of neurons that maximized the classifier's performance, we fit a "random combination ensemble", where we randomly generated 100 different combinations of neurons from the pseudo-population pool. Subsequently, we iteratively trained the model using $n + 1$ neurons from the randomly generated combination.

### Signal pre-processing for local field potentials

Using the FieldTrip toolbox[165] a 4-pole, Butterworth, 250 Hz high pass digital filter was applied to the raw 30 kHz broadband signal. The data were then downsampled to 1 kHz. To eliminate low-frequency noise a 1 Hz high pass filter was applied and a high-frequency artifact rejection algorithm was employed where the data were bandpass filtered between 100 and 250 Hz (9-order Butterworth filter and boxcar of 0.2) and the Hilbert envelope of the signal was calculated and subsequently Z-scored. The artifact threshold was defined at 4 Z-score and a window including 100 ms before and after threshold crossing was rejected.

### Time–frequency representation

To calculate the oscillatory power of the LFP and its time–frequency representation, we used Morlet wavelets, implemented with the FieldTrip toolbox. The wavelets consisted of 7 oscillations, with center frequencies ranging from 1 to 30 Hz (in even steps of 1 Hz), and a 25 ms smoothing time window. Baseline normalization was implemented to account for $1/f$ activity using the decibel conversion method, described as follows:

$$dB = 10 \log_{10} \left( \frac{\text{signal}}{\text{baseline}} \right) \qquad (16)$$

where the baseline was estimated as the fractal component from the power spectrum calculated using the Fitting Oscillations and One-Over-F (FOOOF) algorithm[166] (FieldTrip implementation).

### LFP theta phase modulation

We used MATLAB's *butter* function to design a 4th-order Butterworth bandpass filter, to isolate the theta band LFP signal (MATLAB's *filtfilt* function, zero-phase filtering). Following bandpass filtering, the Hilbert transform was applied to compute the analytic signal, which was then used to derive the instantaneous phase. We assessed significance of theta phase clustering by conducting a Rayleigh test (Toolbox for circular statistics with MATLAB[167]) around a 400 ms window centered around rapid head movement's peak velocity. Critical values for Rayleigh's Z were determined on a significance level of 0.01.

### Neuronal modulation to head movements

Our classification of modulated units builds upon the framework presented in the work of a previous study[91]. However, our approach differs in that we introduce an additional category of modulation, namely units that are both up and downmodulated.

Spike timestamps for a given single neuron are exported from the Plexon software as binary spike trains with a 1 ms time resolution. We extracted spike train epochs (a window of 400 ms) centered on the peak velocity of head movements and divided that window into 8 equally timed bins (50 ms each). We calculated average spike rates across all head movements for each bin. Next, we generated a shuffled distribution of firing rates by circularly permutating spike times, repeating this process 1000 times. Cells were classified as upmodulated if the real average firing rate exceeded the 97.5th percentile of the shuffle average firing rate at any bin and across bins (shuffle distribution of maximum firing rates across the entire 400 ms window). Cells were classified as downmodulated if the real average firing rate was lower than the 2.5th percentile of the shuffle at any bin and across bins (shuffle distribution of minimum firing rates across the entire 400 ms window). Cells were classified as down/upmodulated if their real average firing rate both exceeded the 97.5th percentile and was lower than the 2.5th percentile of the shuffle at different time bins.

### Reporting summary

Further information on research design is available in the Nature Portfolio Reporting Summary linked to this article.

## Data availability

Source data are provided with this paper, and can be downloaded at https://doi.org/10.6084/m9.figshare.25052960. The complete dataset used in this study is available upon request to D.B.P. or J.C.M.T.

## Code availability

The code used to analyze the data in this manuscript has been made available on GitHub at https://github.com/DiegoPiza/3Dmarms[168].

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

## Acknowledgements

We thank registered veterinary technicians Kim Thomaes and Kristy Gibbs from Western University for technical assistance in surgeries and animal care; Kevin Barker from Neuronitek for design and manufacturing of implantable recording chambers and experimental setup; Jonathan C. Lau from the Division of Neurosurgery, Western University for providing advice in neuro-navigation and targeting of deep brain structures; Stephen Frey from Rogue Research, Montreal, Canada for providing technical advice regarding microelectrode array implantation, neuro-navigation and manufacturing of custom microdrives; Megan Roussy from the Natural Sciences and Engineering Research Council of Canada (NSERC) for providing advice regarding spike sorting and data analysis; John Reynolds from the Salk Institute and Zachary Davis from the University of Utah for providing advice regarding initial behavioral training of the marmosets; Gustavo Morrone Parfitt from Genentech, SF, CA, Borna Mahmoudian and Vaishnavi Sukumar for providing assistance during surgery and data collection; Joseph Umoh from Western University, for providing assistance with micro-CT imaging; Miranda Bellyou from Western University for providing assistance with 9.4T MRI imaging. This work was supported by the Canadian Institute of Health Research Project Grant (CIHR); Natural Sciences and Engineering Research Council of Canada (NSERC); Provincial Endowed Academic Chair in Autism; Canada Foundation for Innovation (CFI); Western University BrainsCAN award grant and Healthy Brains, Healthy Lives (HBHL).

## Author contributions

D.B.P., J.C.M.T., S.C. and C.C. designed and planned the study. D.B.P. and J.C.M.T. planned and performed the surgeries. D.B.P. collected, preprocessed and analyzed the data and created the figures for this manuscript. R.A.G. developed code to quantify the SIC and provided expert knowledge guiding the data analysis and writing of this manuscript. B.W.C. developed code to quantify and classify eye movements, quantify the BI and provided expert knowledge to guide the data analysis and writing of this manuscript. L.M. provided expert knowledge to guide the analysis of time/frequency response in the LFP data. D.B.P. and J.C.M.T. wrote and edited this manuscript.

## Competing interests

The authors declare no competing interests.
