## [Peer Review File · Nature Communications]

REVIEWER COMMENTS

Reviewer #1 (Remarks to the Author):

This manuscript describes a study of visual representations of space in the hippocampus of freely-moving marmosets. Piza and colleagues record the activity of neurons in CA1 and CA3 as the monkeys move through a 3D environment retrieving food from dispensers and analyzed subjects' gaze directions and physical location relative to hippocampal activity, including parsing pyramidal and interneurons. They authors also used an existing database in rats to compare head/body movements relative to gaze in the marmosets. They conclude that marmoset hippocampus encodes visual space but only limited responses to physical location, in contrast to rodents, and that different cell types contribute to different parts of the visual spatial representation. Overall, this is an important and exciting study with several novel results that are likely to be of interest to a broad scientific community. I do have several questions that are intended to help clarify some issues that I think will help to strengthen the paper.

- How was spatial selectivity contrasted with occupancy? Classically 'place cells' are determined by an animal moving through space at a certain velocity (20cm/s) and a similar metric should be applied for vision. There is a difference between a neuron responding when an animal is focused on a location (occupancy) versus visually scanning through that location, as the latter would more closely align with encoding self-position. The spatial information content analysis would suggest that occupancy was the primary determinant but given that not all places in space are going to be sampled equally by each animal each session, a location determined to be spatially selective could just simply be any location the animal looked at a lot or sat at a lot, so more spikes would happen in those locations. In this way, decoding could just tell you where the animal liked to look, and not about spatial encoding. I presume the authors controlled for this, but precisely how is not evident so some explanation would be helpful. And if the assessment of spatial selectivity is based on occupancy (after proper controls) and not visual or physical travel through that space, then this difference should be emphasized as it is quite different than conventions in rodents.

- What is the relationship between meaningful features of the environment and view selectivity? In other words, are the view selective targets the locations food dispensers or doors in/out of the environment? If so, how do you distinguish between neurons responding because of memory of that location's significance and more general spatial selectivity? For example, in Figure 3e seems to be the location of the Entrance, and Figure 3f of a dispenser. If this is always the case, then it is important to elaborate on this issue further, particularly if as described above responses are driven only when the animal looks at those locations and not as the gaze travels through it. If a neuron fires when an animal looks at a food dispenser, how do you determine that this is spatial and not some other memory response. While it is certainly interesting that the same neuron fires when the animal looks at that

location from different directions, and that the neuron is not driven when looking at all dispensers, that is not the same as being spatially selective if the response is more driven by the memory of the reward provided at that location, or a combination of reward x location.

- Please add tick marks to the abscissa and ordinate axes in all plots that contain tick labels. Furthermore, please list the bin width in the caption corresponding to each histogram. This is imperative if the reader wants to calculate something using a y-axis that just says “Probability” (Fig. S1D). Finally, please add reasonable scale bars to all figure panels representing physical space— for example, Fig. S1G-J and Fig. 3A,D-F.

- An additional general concern is the apparent lack of a code release. I would like to test your models myself, and I shouldn’t have to wait until the authors eventually see and respond to an email. Furthermore, I would like to see something polished, which is not always delivered in response to such an email. Consider using a data repository that provides a doi such as Dryad (which was free for our institution). Consider including human-readable instructions for opening and running your code. Feel free to store your polished, code release on GitHub for the benefit of your coding résumé. Please generate a few key figures, such as Fig. S3B-C, starting from the data, training any relevant models, and plotting any model results against the data.

- On page 2, “A few studies in freely moving monkeys^{14,30–32}” What kinds of monkeys are referenced in these references? References 30-32 just say “monkeys” in the title, and it’s impossible to tell which monkeys without reading the References. I presume it is rhesus macaques, but please specify to make it clear to the reader.

- On page 8, “For subject C, out of all the sampled view bins,” Please state how many view bins there were in total for each subject C and P.

- On page 8-9, “In contrast, of all visited place bins, in both subjects” Please state how many place bins there were in total for each subject C and P.

- In Fig. 4A, please indicate any statistically significant differences in median values using the appropriate Wilcoxon-Mann-Whitney test.

- In Fig. 5E, please discuss the meaning of the solid red line in the appropriate location of the figure caption.

- On page 12, “We fit a Naka-Rushton function to the decoder performance as a function of ensemble size and calculated statistics on the fit coefficients.” Please state how you fitted the Naka-Rushton function.
- In Fig. 7A,D please add the R^2 annotation corresponding to the cyan trace. Also, please add color bar label to Fig. 7C,F. Furthermore, please show the data points used to fit these Naka-Rushton functions in Fig. 7A,D — even if it looks messy, at least show these data points with the fits in the Supplement.
- On page 16, “The differences in activation latencies and the proportion of the different modulation types suggest that phase-resetting may be causally linked to interneuron activation that is followed by a range of modulation in pyramidal cells.” Please provide some brief reasoning and/or citations for your suggestion.
- In Fig. 8A-C, please add a vertical, solid line to indicate the 0 point. Furthermore, please indicate time points with a significant difference from baseline in Fig. 8A-C. Furthermore, in Fig. 8C, please provide the number of neurons N considered in either population average.
- On page 18, “These differences may have impacted the physiological mechanisms of spatial navigation and specializations in the hippocampus of the two species.” This is an excellent result. Can you show lack of the Theta phase resetting in the saccadic head (or eye) movements of rat?
- On page 18, “‘Bonafide’ place cells as reported in rodents were not found.” Similar to what is mentioned above, please provide the criteria you used in defining “Bonafide” place cells. And similarly, if none are observed, is this due to the navigational behavior of the monkeys? How many times did they travel through a position in space at the necessary speed? Please provide analyses showing that this is not a sampling issue. Without that, the above statement is not justified.
- On page 19, “[...] rather than on maps of space [sic.]” Please fix any typos such as this.
- On page 20, “Thus, mixed selectivity may be the norm in hippocampus neurons.” Please add a brief description of your reasoning. There are published papers in rodents that make this point, as well as the recent Mao et al. 2021 Neuron paper so integrating this point with the broader literature would be helpful

- On page 21, “Here we report an LFP modulation described as head-gaze theta phase-resetting^{22,37,84,123,124}.” Please show evidence that the theta phase is actually reset at the 0 point. So far, you only show firing rate is modulated, which is not the same as theta phase being reset.

The following regard the Supplemental Materials.

- On page 23, “Reward locations were placed at 4 different positions [...]”. Please state how many reward locations there were in total.
- On page 24, please replace “um” with “ μm ” throughout the Supplement. Many keyboards can easily access the symbol “ μ ” using option+m. Furthermore, please capitalize “Hz” instead of “hz” on page 27.
- On page 25, “Quaternions offer an advantage over traditional Euler’s axis angles in that they are a more efficient way to compute rotations and are not susceptible to gimbal lock.” Please add discussion of what gimbal lock means and how it can be avoided with quaternions (e.g. $\text{SO}(3)$ is simply connected).
- On page 27, “Where the proportion of occupied time in each spatial bin is (P_i) and λ is the mean firing rate.” Please define the remaining symbols used in computing the SIC, λ bar and the λ eyes.
- On page 27, “Speed score higher than 0.3” Please explain how this threshold was determined. Even if it is just 53,69.
- Please center align all equations.
- Furthermore, on page 27, please define the x eyes and the y eyes when you write, “the time series of speed and the time series of firing rate.”
- On page 28, “As reported in the results section, [...]”. Please provide a figure reference at the end of this clause. This is for the reader who starts by reading the Supplement, and needs a place to go in the main text to verify what is being stated in the consequent clause.

- On page 28, “This null distribution was obtained by randomly circularly shifting the vectorized firing rate a total of 50 times.” Please state how this number was determined. Do any key results change if this number is increased?
- On page 28, “To establish statistical significance, we compared the actual explained deviance (ED) with the 95th percentile of the null distribution (Bonferroni corrected).” Please provide your formula for ED and define all necessary variables.
- On page 28-29, “The training process concluded when the incremental improvement in model fit by adding more variables became statistically negligible.” Please add your formula for incremental improvement and define all variables used. Furthermore, please add your threshold for statistically negligible.
- On page 29, “These predictions are then compared to the actual neuron's rasters using cross-validation with 5 folds.” Please specify whether your cross-validation was uniform or stratified.

Reviewer #2 (Remarks to the Author):

The paper by Piza et al, is the first one ever to document functional properties of hippocampal cells in freely moving primates in 3D, that are totally able to move, walk and climb freely in all 3 directions. Previous papers on spatial selectivity in non-human primates, documented hippocampal activity in one 1D (alley like enclosures) , 2D (open field like enclosure) or virtual reality. By recording in freely moving marmosets exploring a 3 levels environment akin to the marmoset’s natural ecological behavior, exploiting 3D, the authors discovered a majority of neurons in the hippocampus exhibited a selectivity for view, or mixed selectivity for view and head direction or place. They found no bona fide place cell like activity. The study is extremely well conducted and via a cross species comparison of behavioral locomotion during exploration that is unprecedented, the study offers a phylogenetical framework in which to interpret the results relative to the neural activity observed in the marmoset. The authors suggest that the great differences in neural selectivity may stem from important differences in the nature of explorative behavior between rodents and primate. Marmosets tended to move their head during periods of rest while during locomotion, their head appeared centered. Rodents exhibited a opposite pattern. Overall, the findings are extremely important given the discrepancy of the results with respect to place selectivity identified in rodents. Given the importance of identifying functional properties of hippocampal cells with respect to the processing of space across species, this manuscript is very timely and provides a significant and conceptual advance to the field. I have a few remarks and comments.

1) There is no information relative to the degree of familiarity with the enclosures in marmosets or rodents. This may impact explorative behavior in either species. Can the authors provide some information about this in the text, especially since in the rats, exploration of a novel environment may lead to a more active locomotion.

2) The number of bins used to compute the information content influences the value of the SIC. Therefore, IC computed for view (106bins) or place (84 bins) can not be directly compared unless the number of bins is somehow equalized. I don't think z-scoring solved the problem given that the view values may be overall inflated given the higher number of bins. This is not a deep issue for the general interpretation of the results because the GAM allows to compare predictor strength, but direct comparisons of the SIC should be avoided.

3) In head-gaze shifts, the eye movements generally end before the head movement, as the gaze reached its target before the head. Can this impact the results in any way? Can the authors discuss this in terms of latency if relevant?

4) It is unclear how decorrelated variables are at the behavioral level. Can the author provide some information about the samples of head direction or view with respect to position? In other words, did the animals travel some paths in a multidirectional way, or was their trajectories unidirectional?

5) it looks like animal's facing location is always assumed to be on the maze's wall, while, judging by the video, gaze seemed sometimes to be within the maze which seems to require more visual inspection given that animals move within the maze. How did the authors deal with this?

5) it is unclear to me whether the cells used to construct the pseudo population were recorded in the same session or in different sessions? I understand that place can be decoded from these non-random populations, in which neurons exhibit some selectivity which covaries with place. I am then not sure how to interpret this. For example, time could be decoded from population of premotor neurons recorded during fast or slow arm movements, but it does not mean that the neurons encode time, just that one can decode time from movement. Maybe the point is precisely to say that the position decoding is a by product from other selectivity? the position resolution is quite crude (bin width of 30x 60cm) compared to what is generally achieved in the rodent. This should be stated clearly that while some position decoding can be achieved, it does not achieve the resolution normally found in rodents (<5cm).

6) some reference to work characterizing hippocampal representation of volumetric space in rodents (Jeffery's lab) should be made, especially since this is the only other work examining representation of volumetric space in non-flying animals. This provides a comparative for cells selectivity in rodents in very similar set-up.

7) the reference to Baraduc et al., should be swapped to rather cite Wirth et al., 2017 Plos Biology for accuracy given citing context.

Reviewer #3 (Remarks to the Author):

Primacy of vision shapes behavioral strategies and neural substrates of spatial navigation in the hippocampus of the common marmoset

This is an interesting paper. Part of the interest is that whereas most primates can use eye movements to view different parts of scenes, marmosets use head movements, with a rather limited range of different eye positions. I have the following suggestions to help the authors improve the paper.

1. More attention should be paid to a balanced design for analysis of the extent to which hippocampal neurons respond to view or place. The point has been made that it is very important to provide analyses in which the same views are seen from the same set of places, and these analyses have been performed for macaques. This point and the previous evidence from other primates is made for example in the following papers:

Rolls, E. T. (2023) Hippocampal spatial view cells for memory and navigation, and their underlying connectivity in humans. *Hippocampus* 33: 533-572. doi: 10.1002/hipo.23467.

Georges-François, P., Rolls, E.T. and Robertson, R.G. (1999) Spatial view cells in the primate hippocampus: allocentric view not head direction or eye position or place. *Cerebral Cortex* 9: 197-212.

Rolls, E.T., Treves, A., Robertson, R.G., Georges-François, P. and Panzeri, S. (1998) Information about spatial view in an ensemble of primate hippocampal cells. *Journal of Neurophysiology* 79: 1797-1813.

The point here is that if data are included in an analysis in which only some views can be seen from some places, then any analysis of whether the encoding is for view or place has confounds. If this cannot be addressed, the point should be made that this would be important for future research in marmosets, and that evidence of this type is already present for macaques.

Mixed selectivity for place vs view will be incorrectly inferred unless a balanced design for place x view analysis is used. That point was made in the following paper, and as this is so important for future research, this should be clearly emphasised.

Rolls, E. T. (2023) Hippocampal spatial view cells, place cells, and concept cells: view representations. *Hippocampus* 33: 667-687. doi: 10.1002/hipo.23536.

2. In the section on coding of speed by single neurons in freely moving marmosets, the authors should make it clear that a population of neurons in the macaque hippocampus codes for whole body motion, with different neurons coding for linear and angular velocity, and some neurons providing evidence for inputs from the vestibular system, and others from optic flow.

O'Mara,S.M., Rolls,E.T., Berthoz,A. and Kesner,R.P. (1994) Neurons responding to whole-body motion in the primate hippocampus. *Journal of Neuroscience* 14: 6511-6523.

3. The authors could clarify if theta frequency is related to the amplitude of head movements, or to the speed of movements in the environment. In rats, theta frequency was reported to increase with the height of a jump that was being made.

4. The paper would be strengthened if evidence was available from eye position measured during the recordings. Can any evidence by now be included on this?

5. Further, the paper would also be strengthened if recording data was available from more marmosets.

6. The legend to Fig. 3 could be improved by describing in more detail exactly what is represented in the different parts of parts d, e and f.

7. The video made available on YouTube was helpful, but the paper would be greatly improved if the views present from different places in the maze could be provided, perhaps in the Supp Mat. It would be helpful to be shown exactly what the marmoset would see of what is outside the maze from different places in the maze. Was any attempt made to rotate the maze in the room?

I hope that these points are helpful to the authors in revising the paper.

Edmund Rolls

REVIEWER COMMENTS

Reviewer #1 (Remarks to the Author):

This manuscript describes a study of visual representations of space in the hippocampus of freely-moving marmosets. Piza and colleagues record the activity of neurons in CA1 and CA3 as the monkeys move through a 3D environment retrieving food from dispensers and analyzed subjects' gaze directions and physical location relative to hippocampal activity, including parsing pyramidal and interneurons. They authors also used an existing database in rats to compare head/body movements relative to gaze in the marmosets. They conclude that marmoset hippocampus encodes visual space but only limited responses to physical location, in contrast to rodents, and that different cell types contribute to different parts of the visual spatial representation. Overall, this is an important and exciting study with several novel results that are likely to be of interest to a broad scientific community. I do have several questions that are intended to help clarify some issues that I think will help to strengthen the paper.

We greatly thank the reviewer for the thorough assessment of this manuscript, the positive feedback, and the highly constructive comments. We particularly value the reviewer's attention to detail and identification of important methodological aspects not previously addressed in the manuscript, which have now been incorporated. We have added further analysis and discussion points to clarify and strengthen our claims and implemented changes to the text and figures that were suggested. Below we address each point individually.

1)• How was spatial selectivity contrasted with occupancy? Classically 'place cells' are determined by an animal moving through space at a certain velocity (20cm/s) and a similar metric should be applied for vision. There is a difference between a neuron responding when an animal is focused on a location (occupancy) versus visually scanning through that location, as the latter would more closely align with encoding self-position. The spatial information content analysis would suggest that occupancy was the primary determinant but given that not all places in space are going to be sampled equally by each animal each session, a location determined to be spatially selective could just simply be any location the animal looked at a lot or sat at a lot, so more spikes would happen in those locations. In this way, decoding could just tell you where the animal liked to look, and not about spatial encoding. I presume the authors controlled for this, but precisely how is not evident so some explanation would be helpful. And if the assessment of spatial selectivity is based on occupancy (after proper controls) and not visual or physical travel through that space, then this difference should be emphasized as it is quite different than conventions in rodents.

- The reviewer makes some important observations. We will try to organize our response by addressing one issue at a time.

Determining SIC during body movements and stops.

Indeed, classical rodent place cells are defined during epochs of active movement. In our preliminary data analyses, we initially defined place selectivity with conventional criteria, and only movement epochs were included (see **Response Figure 1a**, which we now include in the supplementary material in Fig. S4a). While the median spatial information content (SIC) between movement and all epochs was not significantly different (two-sided Wilcoxon signed-rank test, $p=0.098$, $Z=1.65$), the percentage of place cells was reduced when analyzing only movement epochs (32% all epochs to 24% only movement epochs). When contrasted with the rodent literature this seems contradictory. This result may have several explanations. One straightforward explanation, as the reviewer suggests, is that because the view and place selectivity interact, this difference might be due to not including sufficient epochs with different views when the animal was moving relative to when it was stationary, as demonstrated in the main manuscript in figure 1 most head movement exploration occurs during body stops. For this reason, we included all epochs in the analyses (see below).

Response figure 1. a) Distribution of spatial information content (SIC) during all epochs, movement epochs, and shuffled controls. **b)** Distribution of inter-event interval durations for eye saccades measured during chaired head fixation (green) and head movements during freely moving experiments in the maze (blue), bin width = 100ms.

Sampling of locations in the maze and controlling for occupancy

We demonstrated a high degree of mixed selectivity of place and view, so we decided to ultimately include all epochs in the analyses of the main figures, with the objectives of comparing the same epochs for both view and place models (as similarly done in (Mao et al., 2021) and obtaining the largest possible number of sampled locations for a given view, and vice versa. Additionally, when computing SIC and generalized additive models (GAM), we control for high occupancy bins by dividing the spike count by the time spent at that location. For both SIC and GAM analyses, we ran circular shifting shuffle controls.

That said we must acknowledge that in our task and many real-world tasks with primates, it is difficult to obtain a homogenous sampling of the maze. Primates use visual exploration to locate rewards (far sensing). In our paradigm, we had different reward locations, and the reward was intentionally proportionally distributed amongst them (e.g., the number of trials where a location was rewarded). This allowed us to control for the reward bias, at least for a fixed number of locations. We have previously shown that by fixing reward locations in a virtual arena in macaques compared to a random reward location schedule, hippocampal neurons changed their place selectivity (Gulli et al., 2020) suggesting this is not a sampling issue but has to do with multidimensional encoding of object/location. We have also shown view encoding in the macaque hippocampus in the same virtual maze (Corrigan et al., 2023). In our 3D maze, it was difficult to do the same since manipulations in the real world are more difficult to implement than in the virtual environment. However, we did corroborate that every view field from all view selective cells was sampled from at least 3 different locations (see **Response figure 2a**). In fact, we found that most view fields were sampled from 10+ unique locations. These unique locations were spread apart from each other as far as 100cm (**Response figure 2b**) and were viewed from different angles (60° +, **Response figure 2c**).

Response figure 2. a) Distribution of viewpoints from unique places per every view field. b) Greatest Euclidean 3D distance between the unique sampled places per view field. c) Greatest angular distance between view projections sampled from unique places per view field.

We have added a clarification to this in the main text (see page 19 line 574)

Head movement sampling

The reviewer raises the point that sampling head movements may be similar to sampling place through translations in the maze. This is a complicated issue and behavioral comparisons of physical translation movements and rapid head movements (or gaze shifts), may not be suitable using the same metrics. As illustrated in **figure 1c** of the main text, marmosets' eye and head movement patterns are kinematically stereotypical (as depicted in the main sequence). Head exploratory movements in the marmoset are ballistic, very fast, and last ~50-200ms depending on amplitude, their main purpose is to rapidly shift the view to a different location rather than to smoothly visually sample locations during the movement. Rapid head movements (we colloquially call head saccades), akin to eye saccades, have a mean inter-event interval of roughly 200-250 ms, with over 75% of intervals being less than 1 second and over 90% less than 2 seconds (see **Response figure 1c**). Because of all the above, we suspect that saccadic suppression a widely documented phenomenon in the macaque visual cortex (Berman et al., 2017), and recently documented during saccades in head-restrained marmosets (Kaneko et al., 2022), is happening during a rapid head movement, which is preceded by a saccade. This strongly suggests that active sampling of view occurs during gaze fixations, rather than during the movement. We are expanding on this issue in a separate manuscript since it is extensive and needs more analyses. Hope the reviewer understands. We now comment on this issue in the manuscript (see page 20, line 588).

- What is the relationship between meaningful features of the environment and view selectivity? In other words, are the view selective targets the locations food dispensers or doors in/out of the environment? If so, how do you distinguish between neurons responding because of memory of that location's significance and more general spatial selectivity? For example, in Figure 3e seems to be the location of the Entrance, and Figure 3f of a dispenser. If this is always the case, then it is important to elaborate on this issue further, particularly if as described above responses are driven only when the animal looks at those locations and not as the gaze travels through it. If a neuron fires when an animal looks at a food dispenser, how do you determine that this is spatial and not some other memory response. While it is certainly interesting that the same neuron fires when the animal looks at that location from different directions, and that the neuron is not driven when looking at all dispensers, that is not the same as being spatially selective if the response is more driven by the memory of the reward provided at that location, or a combination of reward x location.

- The reviewer raises an important and complex issue. Distinguishing whether view-selectivity corresponds to locations in space being looked at independent of the presence of objects such as reward dispensers, entry/exit points, and other relevant environmental cues remains a fundamental question in the field. SIC analysis of view separating rapid head movement epochs from no head movement (fixation) epochs shows that indeed view selectivity is stronger during the absence of head movements (i.e., during head fixations), as shown in **Response Figure 3** (now added to the supplementary material), median SIC is higher when there are fewer head movements (two-sided Wilcoxon signed-rank test, $p=8.29e-15$, $Z=7.76$), which may suggest neurons are responding to something being looked at. This raises questions regarding whether this response stems from memory, object, or purely spatial encoding. However, the latter is a complicated concept because in general monkeys and humans do not look at empty space, there is always "something there". This is why the concept of view arises, as a configuration of spatial cues that can be used to "encode" or identify a location in space. We did several things to minimize the issue of view

vs. specific object selectivity. We covered the walls of the room with black curtains to control for a set of cues at least partially. However, there were still nearby cues related to the maze, for example, reward locations. We shall mention that the objects, small tubes, and solenoids at the reward locations were identical. We reason that if the selectivity had been for the reward device, we would have obtained view fields at all reward locations and in the two walls. We did not observe that. We also had two entry ports in the maze, so this also applies. So, it seems that at least for many view fields, the selectivity was not driven by the object alone, but by a combination of objects and location, defined as view. That said, we admit that for some objects, we cannot control for selectivity, in fact, a recent paper has reported category-selective neurons in the marmoset hippocampus (Tyree et al., 2023). Now we clarify in the text that some of the observed spatial selectivity may be due to the interaction of object selectivity and view to different degrees (see page 20, line 589)

Response figure 3 Distribution of spatial information content (SIC) during epochs of low head movement and epochs of rapid head movement and shuffled controls.

- Please add tick marks to the abscissa and ordinate axes in all plots that contain tick labels. Furthermore, please list the bin width in the caption corresponding to each histogram. This is imperative if the reader wants to calculate something using a y-axis that just says "Probability" (Fig. S1D). Finally, please add reasonable scale bars to all figure panels representing physical space— for example, Fig. S1G-J and Fig. 3A,D-F.

Thanks for the suggestions. This has been corrected

- An additional general concern is the apparent lack of a code release. I would like to test your models myself, and I shouldn't have to wait until the authors eventually see and respond to an email. Furthermore, I would like to see something polished, which is not always delivered in response to such an email. Consider using a data repository that provides a doi such as Dryad (which was free for our institution). Consider including human-readable instructions for opening and running your code. Feel free to store your polished, code release on GitHub for the benefit of your coding résumé. Please generate a few key figures, such as Fig. S3B-C, starting from the data, training any relevant models, and plotting any model results against the data.

We apologize for this oversight, a code release was included along with the journal's submission documents for the initial review process but we didn't include the link in the main text, the code is available on GitHub in this repository: <https://github.com/DiegoPiza/3Dmarms> . This link is now referenced in the main text

- On page 2, "A few studies in freely moving monkeys14,30–32" What kinds of monkeys are referenced in these references? References 30-32 just say "monkeys" in the title, and it's impossible to tell which monkeys without reading the References. I presume it is rhesus macaques, but please specify to make it clear to the reader.

Thank you for pointing this out, we've now clarified this in the main text (page 2 line 62). We referred to rhesus macaques and squirrel monkeys.

- On page 8, “For subject C, out of all the sampled view bins,” Please state how many view bins there were in total for each subject C and P.

We’ve now clarified this in the main text, both animals sampled all 106 view bins at least once across all sessions.

- On page 8-9, “In contrast, of all visited place bins, in both subjects” Please state how many place bins there were in total for each subject C and P.

We’ve now clarified this in the main text, both animals sampled all 84 place bins at least once across all sessions.

- In Fig. 4A, please indicate any statistically significant differences in median values using the appropriate Wilcoxon-Mann-Whitney test.

Figure has been updated and the statistic test results have been added to the supplementary statistics table under Fig.4a.

- In Fig. 5E, please discuss the meaning of the solid red line in the appropriate location of the figure caption.

Thanks for pointing out the need for clarifying this, we’ve added the description on the figure caption “Solid red line is a line of slope =1, corresponding dots below it indicates higher AHV than TS, dots above it indicates the opposite”

- On page 12, “We fit a Naka-Rushton function to the decoder performance as a function of ensemble size and calculated statistics on the fit coefficients.” Please state how you fitted the Naka-Rushton function.

The methodology and equation have been added to the methods section (page 26, line 849).

- In Fig. 7A,D please add the R² annotation corresponding to the cyan trace. Also, please add color bar label to Fig. 7C,F. Furthermore, please show the data points used to fit these Naka-Rushton functions in Fig. 7A,D — even if it looks messy, at least show these data points with the fits in the Supplement.

The cyan trace wasn’t fit to any functions which is the reason why there is no R² to report, it corresponds to the mean and 95% confidence interval of the decoding accuracy of a randomized combination of neurons (100 iterations). The raw mean and 95% confidence interval of the 7A,D plots can be found in the supplementary material under **Fig. S8a,b**.

- On page 16, “The differences in activation latencies and the proportion of the different modulation types suggest that phase-resetting may be causally linked to interneuron activation that is followed by a range of modulation in pyramidal cells.” Please provide some brief reasoning and/or citations for your suggestion.

We have now provided appropriate references supporting this claim in the main text (page 16 line 446) “(Crapse & Sommer, 2008b, 2008a; Katz et al., 2022; J. Martinez-Trujillo, 2022)”.

- In Fig. 8A-C, please add a vertical, solid line to indicate the 0 point. Furthermore, please indicate time points with a significant difference from baseline in Fig. 8A-C. Furthermore, in Fig. 8C, please provide the number of neurons N considered in either population average.

The figure has been updated

- On page 18, “These differences may have impacted the physiological mechanisms of spatial navigation and specializations in the hippocampus of the two species.” This is an excellent result. Can you show lack of the Theta phase resetting in the saccadic head (or eye) movements of rat?

This is an excellent observation. Unfortunately, we don’t have the data in a sufficiently controlled manner to answer this question. Although we had the data from Buzsaki’s lab. We would need to have documentation of the environment, illumination conditions, visual cues, etc to know whether the animals were making head movements to visual targets or simply re-allocating the whiskers and orienting the nose. We feel would be unfair and poorly controlled for us to conduct this comparison. We must reach out to laboratories working with rats, and we have done so, to get datasets that allow for this comparison. However these data are not available, so these experiments must be designed and conducted in the future. We hope the reviewer understands this limitation. We have clarified this in the main text (see page 18, line 532) “These differences may have impacted the physiological mechanisms of spatial

navigation and specializations in the hippocampus of the two species. Further experimentation using similar behavioral paradigms and readouts (habituation times, mazes, tasks) between both species would be beneficial to systematically bridge this gap."

- On page 18, "'Bonafide' place cells as reported in rodents were not found." Similar to what is mentioned above, please provide the criteria you used in defining "Bonafide" place cells. And similarly, if none are observed, is this due to the navigational behavior of the monkeys? How many times did they travel through a position in space at the necessary speed? Please provide analyses showing that this is not a sampling issue. Without that, the above statement is not justified.

Thanks for raising this point, every spatial bin included in the analyses was required to be independently visited (actively traversed through at a minimum speed of 15 cm/s) at least 3 times per session. We acknowledge that, behaviorally, marmosets tend to explore visually much more than they engage in physical visits to locations, potentially introducing biases in the representation of spatially relevant variables. We recognize the necessity for different experimental controls to ensure a more equal sampling of space and view. In order to improve objectivity we have decided to revise the statement and add further clarification in the main text (see page 19 line 568)

- On page 19, "[...] rather than on maps of space [sic.]" Please fix any typos such as this.

Thanks for catching this, We have corrected it and checked for any more typos in the text.

- On page 20, "Thus, mixed selectivity may be the norm in hippocampus neurons." Please add a brief description of your reasoning. There are published papers in rodents that make this point, as well as the recent Mao et al. 2021 Neuron paper so integrating this point with the broader literature would be helpful

Thanks for this suggestion. Indeed hippocampal mixed selectivity is a relatively old idea, with an abundance of evidence from rodent literature, and more recently in macaques by Mao et al. 2021, we have now elaborated on this and added the necessary citations to better contextualize this claim (see page 21 line 649).

- On page 21, "Here we report an LFP modulation described as head-gaze theta phase-resetting^{22,37,84,123,124}." Please show evidence that the theta phase is actually reset at the 0 point. So far, you only show firing rate is modulated, which is not the same as theta phase being reset.

We thank the reviewer for this important observation. As the reviewer points out, we didn't show evidence of the theta phase being reset in relation to the onset of head movements. We have now expanded upon this in the results section and added the figure below to the supplementary material, showing the re-alignment of theta phase present after the presence of rapid head movements. (see page 15 line 417)

Response figure 4 (a) heatmap of mean phase concentration around a 400ms window centered on head movement peak velocity for an example channel during a recording session (~1hr) recording for $n=2044$ head movements, dotted gray line indicates -120ms, dotted purple line indicates +120ms **(b)** phase distribution during -120ms (gray) and +120ms (purple) time epochs. Green and red lines indicate circular mean and vector length of the gray and purple distribution respectively. **(c)** Mean Rayleigh's Z-statistic for non-uniformity of circular data corresponding to $n=59$ recording sessions for a 400ms window centered on head movement peak velocity, shaded area represents 95% confidence interval. Gray dotted line indicates critical value for significance at $\alpha=0.01$.

The following regard the Supplemental Materials.

- On page 23, "Reward locations were placed at 4 different positions [...]". Please state how many reward locations there were in total.

We apologize for the omission of this important detail, the total amount of reward locations across the maze was 12, we have updated the main text with this (see page 24 line 766).

- On page 24, please replace "um" with "µm" throughout the Supplement. Many keyboards can easily access the symbol "µ" using option+m. Furthermore, please capitalize "Hz" instead of "hz" on page 27.

Corrected

- On page 25, "Quaternions offer an advantage over traditional Euler's axis angles in that they are a more efficient way to compute rotations and are not susceptible to gimbal lock." Please add discussion of what gimbal lock means and how it can be avoided with quaternions (e.g. $SO(3)$ is simply connected).

We have added the following clarification: " In a system with three rotational axes (such as pitch, yaw, and roll), the alignment of two axes restricts the third. Where rotations along the locked axes produce the same effect as rotations

along the combined axis, this is commonly referred to as gimbal lock. Quaternions constructs avoid gimbal lock by representing space as a three dimensional sphere in four-dimensional space where any loop along it's topology can be continuously contracted to a single point." (see page 26, line 829)

- On page 27, "Where the proportion of occupied time in each spatial bin is (P_i) and λ is the mean firing rate." Please define the remaining symbols used in computing the SIC, λ bar and the λ eyes.

We have clarified this in the main text (see page 28 line 923)

- On page 27, "Speed score higher than 0.3" Please explain how this threshold was determined. Even if it is just 53,69.

As described in (Góis & Tort, 2018), the speed threshold of 0.3: "(1) it separated well the large and small peaks in the actual distribution of speed scores ... (2) all absolute speed scores above this threshold were highly significant when compared to surrogate distributions; and (3) we wanted to avoid spurious correlations due to spatial influences such as place field traversals"

Basically, in that study, the threshold for categorizing a cell as a speed cell was a speed score greater than 0.3, as it effectively separated a small distribution of highly speed-correlated cells within the distribution of the population's speed scores, we observed a similar pattern in our dataset, where this threshold segregated a similar subset of highly speed-correlated cells for both AHV and TS (**Response figure 5, a,b**). With the purpose of preserving consistency between studies we choose to report speed cell counts using the same threshold. Alternatively, to determine speed selectivity in our GAM encoding model, a empirically established threshold is computed for significance per every cell (as described in the methods section), which we think complements the speed score analysis. The reasoning behind choosing this threshold has now been added to the main text in the methods section (see page 29, line 941)

Response figure 5 (a) Distribution of speed score (Pearson correlation coefficient between time series of firing rate and speed) for both angular head velocity (AHV) and **(b)** translation speed (TS). Pink line represents smoothed kernel density function, dotted red line represents speed score = 0.3, green arrow represents small peak of highly speed encoding cells.

- Please center align all equations.

Corrected

- Furthermore, on page 27, please define the x eyes and the y eyes when you write, "the time series of speed and the time series of firing rate."

Corrected

- On page 28, "As reported in the results section, [...]". Please provide a figure reference at the end of this clause.

This is for the reader who starts by reading the Supplement, and needs a place to go in the main text to verify what is being stated in the consequent clause.

Reference has been added to the main text (see page 30, line 972)

• On page 28, “This null distribution was obtained by randomly circularly shifting the vectorized firing rate a total of 50 times.” Please state how this number was determined. Do any key results change if this number is increased?

The selection of 50 permutations was primarily limited by our computation capabilities as running all model combinations and shuffle controls can take several hours per neuron. We admit that 50 times it’s not a customary number in the field and might not be sufficient to approximate the true mean of the noise distribution, we have now since repeated the analyses conducting a total of 100 permutations, which we believe it aligns more closely with standard practices in the field (see **response figure 6** comparison below). The updated results don’t significantly alter our claims. We have updated the figures and results with the 100 permutations analyses in the main text.

Response figure 6 Proportion of encoding putative pyramidal cells (upper panels) and putative interneurons (lower panels), for previous 50 permutation shuffle controls (left) and new 100 permutation shuffle controls (right).

• On page 28, “To establish statistical significance, we compared the actual explained deviance (ED) with the 95th percentile of the null distribution (Bonferroni corrected).” Please provide your formula for ED and define all necessary variables.

We’ve added the requested formula to the methods section (see page 30, line 982).

• On page 28-29, “The training process concluded when the incremental improvement in model fit by adding more variables became statistically negligible.” Please add your formula for incremental improvement and define all variables used. Furthermore, please add your threshold for statistically negligible.

We apologize for this omission, we do not have an explicit formula for ‘incremental improvement’, the term is used more qualitatively to convey that further additions of variables do not lead to a statistically significant enhancement in model fit, as determined by the comparison with the null distribution. It’s worth clarifying that a new null distribution is calculated at every iteration of the training process, via randomly shifting the vectorized new added variable. The statistical significance of this model is then determined by comparing the actual explained deviance (ED) with the 95th percentile of the null distribution. This threshold is Bonferroni corrected for multiple comparisons (total amount of models tested per neuron). We’ve clarified this in the main text (see page 31, line 1002)

• On page 29, “These predictions are then compared to the actual neuron’s rasters using cross-validation with 5 folds.” Please specify whether your cross-validation was uniform or stratified.

We thank the reviewer for this observation, initially we had implemented uniform cross-validation. However, recognizing that the data predictors were not uniformly distributed across folds, we deemed more appropriate to use stratified cross-validation. We implemented this change along with the increase in permutations describe in the

response above. We have now updated the methods section with this description (see page 31, line 1010).

Reviewer #2 (Remarks to the Author):

The paper by Piza et al, is the first one ever to document functional properties of hippocampal cells in freely moving primates in 3D, that are totally able to move, walk and climb freely in all 3 directions. Previous papers on spatial selectivity in non-human primates, documented hippocampal activity in one 1D (alley like enclosures) , 2D (open field like enclosure) or virtual reality. By recording in freely moving marmosets exploring a 3 levels environment akin to the marmoset's natural ecological behavior, exploiting 3D, the authors discovered a majority of neurons in the hippocampus exhibited a selectivity for view, or mixed selectivity for view and head direction or place. They found no bona fide place cell like activity. The study is extremely well conducted and via a cross species comparison of behavioral locomotion during exploration that is unprecedented, the study offers a phylogenetical framework in which to interpret the results relative to the neural activity observed in the marmoset. The authors suggest that the great differences in neural selectivity may stem from important differences in the nature of explorative behavior between rodents and primate. Marmosets tended to move their head during periods of rest while during locomotion, their head appeared centered. Rodents exhibited a opposite pattern. Overall, the findings are extremely important given the discrepancy of the results with respect to place selectivity identified in rodents. Given the importance of identifying functional properties of hippocampal cells with respect to the processing of space across species, this manuscript is very timely and provides a significant and conceptual advance to the field. I have a few remarks and comments.

We thank the reviewer for the valuable comments. We appreciate your positive acknowledgment of our work. We believe that your comments and constructive feedback have helped us clarify and refine our claims. We are especially thankful for your remarks regarding correlated behavior and projecting view only to the maze's walls. We try to address each point below.

1) There is no information relative to the degree of familiarity with the enclosures in marmosets or rodents. This may impact explorative behavior in either species. Can the authors provide some information about this in the text, especially since in the rats, exploration of a novel environment may lead to a more active locomotion.

We agree that clarifying this is important, as novelty can dramatically impact exploration patterns. First, it is hard to precisely know how many visits each species requires to become familiar with the maze. However, for the data we are using we believe that both our subject groups (rats and marmosets) were habituated enough to the maze that novelty was likely not an issue. For the rat dataset (full description can be found here <https://crcns.org/files/data/hc2/crcns-hc2-data-description.pdf>) the animals were habituated on the arena at least 3 days before recordings, in our case marmosets were habituated for 2 weeks. Marmosets are highly social and we notice (by observation) that they needed at least two weeks to fully accept reward from experimenters and move to seek it in the maze. Notably, we observed that upon initial exposure to the maze, the marmosets tend to limit active movement and remain at one place in the maze for long periods. This did not look like a freezing, fear-conditioning response. They often selected elevated positions that allowed visual exploration of all levels. We think this is due to the animals minimizing energy spending during exploration. Over time, marmosets learned to recognize reward sites and increased their overall movement when motivated to obtain a reward. We reason they use vision over olfaction, audition, and somatosensorial, while rats may use a different proportion of these senses. We now clarify the habituation periods of the different species (see page 5, line 129).

2) The number of bins used to compute the information content influences the value of the SIC. Therefore, IC computed for view (106bins) or place (84 bins) can not be directly compared unless the number of bins is somehow equalized. I don't think z-scoring solved the problem given that the view values may be overall inflated given the higher number of bins. This is not a deep issue for the general interpretation of the results because the GAM allows to compare predictor strength, but direct comparisons of the SIC should be avoided.

We agree with the reviewer that comparisons between SIC of two variables that had different sampling patterns and total bin numbers should be avoided, as pointed out in your comment, we tried to mitigate this by normalizing (z-scoring) the SIC to the null distribution (shuffle control), but this comparison might not be straightforward, which is the very reason why we decided to use a multivariate analysis (GAM). We have now removed this comparison from the main text (see page 8, line 241).

3) In head-gaze shifts, the eye movements generally end before the head movement, as the gaze reached its target before the head. Can this impact the results in any way? Can the authors discuss this in terms of latency if relevant?

This is an excellent point we have considered extensively due to previous experience we have had in the head-unrestrained macaque. We agree that there is uncertainty in latency regarding head-gaze shifts due to the absence of freely moving eye-tracking data. Previous work on microstimulation of Supplementary Eye Fields (J. C. Martinez-Trujillo et al., 2003) showed that head movements exhibit longer latency (10-15ms) compared to eye movements when initiating gaze shifts. Furthermore, at the end of a head-eye gaze shift, there is a period with a duration ranging from 50-200ms, depending on the amplitude of the gaze shift commonly referred to as the vestibuloocular (VOR) phase, where the eyes remain fixed in place (rolling opposite to head movement direction) while the head continues to move. At least for macaques, at the end of the gaze shift the eyes and the head tend to align, which positions the eye in the head at a kinematically optimal position to initiate another gaze shift. There is of course a range of eye in head positions the eye could assume depending on the task, amplitude of the gaze shifts, and other variables, which makes the topic complex, at least for small gaze shifts (J. C. Martinez-Trujillo et al., 2003).

Regarding relative latencies of the eye and the head unfortunately we cannot provide those measurements, because we did not measure eye movements. It is technically very challenging, particularly in the 3D maze. However, we assume the physiology of marmosets and macaques may be similar, and even if commands arrive at the eye and head/neck muscles at the same time, the lower inertia of the eye will result in shorter latencies than the head. This, however, must be corroborated experimentally. In our analyses, we aligned neural data to head peak velocity because we did not have estimates of the eye latency. We observed modulations of both LFP and firing rate ~120ms before the head peak velocity (**Fig.8** of the main text). So, these data must be interpreted by considering these issues. (**Now we comment on this in page 22, line 688**)

4) It is unclear how decorrelated variables are at the behavioral level. Can the author provide some information about the samples of head direction or view with respect to position? In other words, did the animals travel some paths in a multidirectional way, or was their trajectories unidirectional? Show

The reviewer raises a fundamental and important issue. We have conducted new analyses and prepared a new figure (see below) with data showing that most paths are traveled.

Response figure 7(**a,b**) depicts individual trajectories traveled from example individual sessions (left and center panels) and all sessions (right panel). We considered the departing point of each trajectory as the ($x=0$, $y=0$) point of Euclidean space, so the allocentric heading direction can be traced from that origin. Trajectories span all quadrants and form a quasi-symmetric plot with biases along the longer axis of the rectangular maze indicating that most paths were traveled in complementary heading directions. This was anticipated from our configuration of reward and the structure of our maze. (**c**) shows counts of complementary trajectories traveled along all possible paths between quadrants (SW= southwest, SE= southeast, NW= northwest, NE= northeast) across all trajectories across all sessions. Trajectories head in different directions along each path. This figure is now part of the supplementary material (**Fig. S3a-b**), and has been added to the results section in the main text (**see page 4, line 117**)

Response figure 7 (a,b) individual trajectories traveled from example individual sessions (left and center panels) and all sessions (right panel). Each trajectory originates in the center ($x=0,y=0$) and the heading direction can be traced from that origin **(c)** counts of complementary trajectories traveled along all possible paths between quadrants (SW= southwest, SE= southeast, NW= northwest, NE= northeast) across all trajectories across all sessions (Subject C = 26 sessions, Subject P= 33 sessions).

5) it looks like animal's facing location is always assumed to be on the maze's wall, while, judging by the video, gaze seemed sometimes to be within the maze which seems to require more visual inspection given that animals move within the maze. How did the authors deal with this?

The reviewer points out an important limitation of our view projection analysis. We located the rewards on the walls of the maze so the animals were motivated to look at these locations. However, the maze contains some local cues and landmarks that are situated within the walls, mainly the climbing ropes, climbing platform, and access holes. The animals must have looked at those landmarks. It is possible that for a certain gaze projection, the fixated object was not on the wall exactly but 'along the gaze path' corresponding to that projection. This may add noise to the data, and we may be underestimating view fields by 'pooling' gaze projections that may correspond to different fixation planes. We have attempted to reconstruct some internal landmarks. It is not trivial and requires a full digital reconstruction of the maze's internal objects and geometry, plotting the gaze path and detecting the intersection with the reconstructed objects. To do this precisely requires some measurements using image processing techniques, likely video or laser-guided measurements, and substantial digital work. We hope the reviewer understands this goes beyond the scope of this work. We acknowledge the issue in the text and clarify the limitations of our analyses (see page 20, line 593). We do thank the reviewer for noticing this and it has opened a line of investigation and more precise measurements in our lab.

6) it is unclear to me whether the cells used to construct the pseudo population were recorded in the same session or in different sessions? I understand that place can be decoded from these non-random populations, in which neurons exhibit some selectivity which covaries with place. I am then not sure how to interpret this. For example, time could be decoded from population of premotor neurons recorded during fast or slow arm movements, but it does not mean that the neurons encode time, just that one can decode time from movement. Maybe the point is precisely to say that the position decoding is a by product from other selectivity? the position resolution is quite crude (bin width of 30x 60cm) compared to what is generally achieved in the rodent. This should be stated clearly that while some position decoding can be achieved, it does not achieve the resolution normally found in rodents (<5cm).

The cells were non-simultaneously recorded (we have now clarified that in the main text). Indeed, population decoding of place from rat hippocampal neurons reaches a higher resolution than in our case. As far as we understand it, in order to achieve such high spatial resolution, those ensembles tend to be composed of 60+ neurons (Tampuu et al., 2019; Wilson & McNaughton, 1993), however, lower resolutions of ~30cm can be achieved with ensembles of only 5 neurons (Tampuu et al., 2019). The reason we used such big spatial bins was to ensure we had enough samples for training the model across sessions since spatial sampling was a limitation in our study. As we explained in the answers to reviewer 1, marmosets do not move as much as rats, they use vision to explore the environment and visit locations where a potential or visible reward is present. Our main claim aligns with the reviewer's comments, that place can still be decoded from a population of neurons that are not highly place selective, but mostly mixed selective for place/view/head direction. We have clarified this further in the discussion (see page 21 line 658)

7) some reference to work characterizing hippocampal representation of volumetric space in rodents (Jeffery's lab) should be made, especially since this is the only other work examining representation of volumetric space in non-flying animals. This provides a comparative for cells selectivity in rodents in very similar set-up.

Thank you for the suggestion. The work by (Grieves et al., 2020) is indeed highly relevant to the current study. Specifically, within the context of neural correlates of naturalistic 3D behavior. We admittedly were not comfortable drawing too many parallels with that study because of the differences between their maze and ours. We have now added further context in the main text in reference to this work (see page 19, line 586, page 2 line 45).

8) the reference to Baraduc et al., should be swapped to rather cite Wirth et al., 2017 Plos Biology for accuracy given citing context.

Thank you for bringing this to our attention, we have modified the reference.

Reviewer #3 (Remarks to the Author):

Primacy of vision shapes behavioral strategies and neural substrates of spatial navigation in the hippocampus of the common marmoset

This is an interesting paper. Part of the interest is that whereas most primates can use eye movements to view different parts of scenes, marmosets use head movements, with a rather limited range of different eye positions. I have the following suggestions to help the authors improve the paper.

We thank the reviewer for the comments and suggestions. Below we address each point individually

1. More attention should be paid to a balanced design for analysis of the extent to which hippocampal neurons respond to view or place. The point has been made that it is very important to provide analyses in which the same views are seen from the same set of places, and these analyses have been performed for macaques. This point and the previous evidence from other primates is made for example in the following papers:

Rolls, E. T. (2023) Hippocampal spatial view cells for memory and navigation, and their underlying connectivity in humans. *Hippocampus* 33: 533-572. doi: 10.1002/hipo.23467.

Georges-François, P., Rolls, E. T. and Robertson, R. G. (1999) Spatial view cells in the primate hippocampus: allocentric view not head direction or eye position or place. *Cerebral Cortex* 9: 197-212.

Rolls, E. T., Treves, A., Robertson, R. G., Georges-François, P. and Panzeri, S. (1998) Information about spatial view in an ensemble of primate hippocampal cells. *Journal of Neurophysiology* 79: 1797-1813.

The point here is that if data are included in an analysis in which only some views can be seen from some places, then any analysis of whether the encoding is for view or place has confounds. If this cannot be addressed, the point should be made that this would be important for future research in marmosets, and that evidence of this type is already present for macaques.

Mixed selectivity for place vs view will be incorrectly inferred unless a balanced design for place x view analysis is used. That point was made in the following paper, and as this is so important for future research, this should be clearly emphasised.

Rolls, E. T. (2023) Hippocampal spatial view cells, place cells, and concept cells: view representations. *Hippocampus* 33: 667-687. doi: 10.1002/hipo.23536.

We thank the reviewer for highlighting this important issue. We agree that highly correlated view/place combinations can lead to erroneously classifying some cells as mixed selective, and we apologize for not clarifying this better in the initial manuscript. We took mainly 2 precautions to avoid this issue. **1)** As shown in the figure below, we made sure that every view field from all view selective cells was sampled from at least 3 different locations (see panel **a**). In fact, we found that most view fields were sampled from 10+ unique locations. These unique locations were spread apart from each other as shown in panel **(b)** and were viewed from different angles (60° +). **2)** The nested, stepwise, forward search approach for the GAM models ensures that the variable that better explains neuronal variability gets selected as the best 1st order model, the other variables are fitted next in a stepwise manner, and only if there is statistically significant increment of goodness of fit (compared against a shuffled distribution) is then that variable selected as part of the significant predictors.

Response figure 2 (from response to reviewer # 1) a) Distribution of viewpoints from unique places per every view field. b) Greatest Euclidean 3D distance between the unique sampled places per view field. c) Greatest angular distance between view projections sampled from unique places per view field.

2. In the section on coding of speed by single neurons in freely moving marmosets, the authors should make it clear that a population of neurons in the macaque hippocampus codes for whole body motion, with different neurons coding for linear and angular velocity, and some neurons providing evidence for inputs from the vestibular system, and others from optic flow.

O'Mara, S.M., Rolls, E.T., Berthoz, A. and Kesner, R.P. (1994) Neurons responding to whole-body motion in the primate hippocampus. *Journal of Neuroscience* 14: 6511-6523.

Thanks for pointing this out. Even though we had initially mentioned that “a study found hippocampal cells in macaque monkeys, that responded to linear and rotation-assisted motion (monkey sitting in a remote-controlled robotic platform)” we realize that we did not cite the correct reference when mentioning this finding. We corrected the reference and have now elaborated on this further (see page 20, line 605)

3. The authors could clarify if theta frequency is related to the amplitude of head movements, or to the speed of movements in the environment. In rats, theta frequency was reported to increase with the height of a jump that was being made.

We thank the reviewer for bringing up this issue. Preliminary analyses of the partial correlation between speed and oscillatory power (see response figure 8 below) show a small but significant positive effect of both TS and AHV on theta (TS partial correlation median = 0.024, 95% confidence interval [0.009 0.039]; AHV partial correlation median = 0.047, 95% confidence interval [0.029 0.063]) and alpha power (TS partial correlation median = 0.011, 95% confidence interval [0.004 0.017]; AHV partial correlation median = 0.009, 95% confidence interval [0.003 0.016]), and a not significant modulation of beta power (TS partial correlation median = -0.003, 95% confidence interval [-0.012 0.005]; AHV partial correlation median = 0.003, 95% confidence interval [-0.005 0.01]). However, the values of the correlation coefficients are very low and although they do reach statistical significance, we think these results may be too preliminary to reach firm conclusions. We will expand on this topic in future work.

Response figure 8 Mean partial correlation between oscillatory power (theta, alpha and beta bands) and translation speed and angular head velocity (AHV). Shaded areas (notches of the box plots) correspond to 95% confidence intervals, the median is considered significant if the confidence interval doesn't include the 0 line (gray dashed line).

4. The paper would be strengthened if evidence was available from eye position measured during the recordings. Can any evidence by now be included on this?

Unfortunately, we have a technical limitation to record eye movements in the freely moving set up. The only eye movement recordings we have are head fixed (see Fig.1c, Fig. S1d,e,f), where we show that marmoset saccadic eye movements are within a 5 degrees radius from a central fixation point (Fig. S1f).

5. Further, the paper would also be strengthened if recording data was available from more marmosets.

We agree with the reviewer. Unfortunately, we do not have data from other animals and all the animals have been euthanized. We initially implanted a total of 4 marmosets, but due to the difficulty of successfully targeting the CA regions with deep electrodes, the success rate of surgeries was ~50%, which limited the number of subjects we could include in this work. We believe, however, that the uniqueness of this dataset given the model organism, coupled with the relatively high neuronal sample size (n= 331 single neurons) and the clear and reproducible effects across subjects, provides a robust foundation for our conclusions. This sample size is consistent with previously published work in macaque hippocampal physiology (2 subjects, 183 neurons) (Gulli et al. 2020)

6. The legend to Fig. 3 could be improved by describing in more detail exactly what is represented in the different parts of parts d, e and f.

We have now clarified this legend (see page 8, line 226)

7. The video made available on YouTube was helpful, but the paper would be greatly improved if the views present from different places in the maze could be provided, perhaps in the Supp Mat. It would be helpful to be shown exactly what the marmoset would see of what is outside the maze from different places in the maze. Was any attempt made to rotate the maze in the room?

We have now added the lab views the marmosets saw from the maze (see figure below, now in the main text as **Fig. S2a**), unfortunately we were not able to collect good behavior sessions rotating the maze, as we were trying to optimize the animal sampling space as much as possible, after collecting enough samples of space we would try rotating the maze, and calibrating the motion capture software again (~20-30 minutes), we would find that after that the marmoset did not want to keep exploring. Simplifying the maze would probably help with future attempts at this.

I hope that these points are helpful to the authors in revising the paper.
Edmund Rolls

References

- Berman, R. A., Cavanaugh, J., McAlonan, K., & Wurtz, R. H. (2017). A circuit for saccadic suppression in the primate brain. *Journal of Neurophysiology*, *117*(4), 1720–1735. <https://doi.org/10.1152/jn.00679.2016>
- Corrigan, B. W., Gulli, R. A., Doucet, G., Mahmoudian, B., Abbass, M., Roussy, M., Luna, R., Sachs, A. J., & Martinez-Trujillo, J. C. (2023). View cells in the hippocampus and prefrontal cortex of macaques during virtual navigation. *Hippocampus*, *33*(5), 573–585. <https://doi.org/10.1002/hipo.23534>
- Crapse, T. B., & Sommer, M. A. (2008a). Corollary discharge across the animal kingdom. *Nature Reviews. Neuroscience*, *9*(8), 587–600. <https://doi.org/10.1038/nrn2457>
- Crapse, T. B., & Sommer, M. A. (2008b). Corollary discharge circuits in the primate brain. *Current Opinion in Neurobiology*, *18*(6), 552–557. <https://doi.org/10.1016/j.conb.2008.09.017>
- Góis, Z. H. T. D., & Tort, A. B. L. (2018). Characterizing Speed Cells in the Rat Hippocampus. *Cell Reports*, *25*(7), 1872–1884.e4. <https://doi.org/10.1016/j.celrep.2018.10.054>
- Grievess, R. M., Jedidi-Ayoub, S., Mishchanchuk, K., Liu, A., Renaudineau, S., & Jeffery, K. J. (2020). The place-cell representation of volumetric space in rats. *Nature Communications*, *11*(1), Article 1. <https://doi.org/10.1038/s41467-020-14611-7>
- Gulli, R. A., Duong, L. R., Corrigan, B. W., Doucet, G., Williams, S., Fusi, S., & Martinez-Trujillo, J. C. (2020). Context-dependent representations of objects and space in the primate hippocampus during virtual navigation. *Nature Neuroscience*, *23*(1), 103–112. <https://doi.org/10.1038/s41593-019-0548-3>

- Kaneko, T., Komatsu, M., Yamamori, T., Ichinohe, N., & Okano, H. (2022). Cortical neural dynamics unveil the rhythm of natural visual behavior in marmosets. *Communications Biology*, 5(1), Article 1.
<https://doi.org/10.1038/s42003-022-03052-1>
- Katz, C. N., Schjetnan, A. G. P., Patel, K., Barkley, V., Hoffman, K. L., Kalia, S. K., Duncan, K. D., & Valiante, T. A. (2022). A corollary discharge mediates saccade-related inhibition of single units in mnemonic structures of the human brain. *Current Biology*, 32(14), 3082-3094.e4. <https://doi.org/10.1016/j.cub.2022.06.015>
- Mao, D., Avila, E., Caziot, B., Laurens, J., Dickman, J. D., & Angelaki, D. E. (2021). Spatial modulation of hippocampal activity in freely moving macaques. *Neuron*, 109(21), 3521-3534.e6.
<https://doi.org/10.1016/j.neuron.2021.09.032>
- Martinez-Trujillo, J. (2022). Corollary discharge: Linking saccades and memory circuits in the human brain. *Current Biology: CB*, 32(14), R774–R776. <https://doi.org/10.1016/j.cub.2022.06.006>
- Martinez-Trujillo, J. C., Wang, H., & Crawford, J. D. (2003). Electrical Stimulation of the Supplementary Eye Fields in the Head-Free Macaque Evokes Kinematically Normal Gaze Shifts. *Journal of Neurophysiology*, 89(6), 2961–2974. <https://doi.org/10.1152/jn.01065.2002>
- Parker, P. R. L., Martins, D. M., Leonard, E. S. P., Casey, N. M., Sharp, S. L., Abe, E. T. T., Smear, M. C., Yates, J. L., Mitchell, J. F., & Niell, C. M. (2023). A dynamic sequence of visual processing initiated by gaze shifts. *Nature Neuroscience*, 26(12), Article 12. <https://doi.org/10.1038/s41593-023-01481-7>
- Tampuu, A., Matiisen, T., Ólafsdóttir, H. F., Barry, C., & Vicente, R. (2019). Efficient neural decoding of self-location with a deep recurrent network. *PLoS Computational Biology*, 15(2), e1006822.
<https://doi.org/10.1371/journal.pcbi.1006822>
- Tyree, T. J., Metke, M., & Miller, C. T. (2023). Cross-modal representation of identity in the primate hippocampus. *Science*, 382(6669), 417–423. <https://doi.org/10.1126/science.adf0460>
- Wilson, M. A., & McNaughton, B. L. (1993). Dynamics of the Hippocampal Ensemble Code for Space. *Science*, 261(5124), 1055–1058.

REVIEWERS' COMMENTS

Reviewer #1 (Remarks to the Author):

Piza et al. successfully addressed the overwhelming majority of my concerns, and the manuscript as a whole is greatly improved. I look forward to its eventual publication of this exciting work. I have only a few remaining comments and suggestions that should be readily addressable.

While I agree it is unfortunate that the original authors did not test for lack of Theta phase resetting in rats during head movements, the authors do have the data from Buzsaki's lab for rapid head movements in rats. The main issue is that the Authors cannot determine which head movements are orienting towards a visual target versus which head movements are re-allocation of the whiskers/nose. The Authors can at least average over all sufficiently fast head movements and look to see if you see any clear Theta phase resetting in the available rat data with an appropriate caveat how results may be affected by the consideration of all head movements. For example, the Authors can simply make Fig. S8c-e for the rat data and report how consideration of head movements not directed towards visual targets could be biasing the results towards the non-appearance of Theta phase resetting in rats. I understand that this analysis will be limited in the sense that it will not be able to distinguish between head movements that are and are not directed towards a visual target, but the analogous. Simply make this caveat clear to the reader and show the results, whatever they may be.

For lines 691-692 on page 23, "We observed modulations of both LFP and firing rate ~ 120 ms before the head peak velocity (Fig. 8a-c)". I have a concern regarding the significant activity you observed in the LFP and firing rate traces at time points < 120 ms before peak head velocity. I suspect it is due to modulation from the previous jump in head-gaze direction. Can you show this significant activity still exists when you exclude jumps in head-gaze direction that begin within 120 milliseconds of the end of another jump in head-gaze direction?

For lines 832-834 on page 27, "Quaternion constructs avoid gimbal lock by representing space as a three-dimensional sphere in four-dimensional space where any loop along its topology can be continuously contracted to a single point". This is an excellent addition. That being said, you have a typo ("it's topology" \rightarrow "its topology"). Furthermore, I think it would benefit the reader to include some more physical intuition here. For example, you could add how equivalently, rotations in three-dimensional space are simply connected. On a separate note, were the real components of your quaternionic representations nonzero? The most common way to represent rotations with quaternions are purely imaginary, with the real q_0 term reserved for the exponent of scale transformations. Hence, $q_0=0$ corresponds to the trivial transformation of scale. If indeed $q_0=0$, it would be good to mention that to the reader, as well, along with what it means in terms of trivial scale transformations.

On lines 847-848 on page 27, “Movements with speeds higher than 300cm/s were deemed artifactual [sic.]”. You are missing a period here.

Of lesser importance, I noticed a minor error in your code release. The error is on line 143 of ‘3Dmarms/Speed Score/Speed_Score_demo.m’. The problem only manifests if you change the number of shuffles, numofshuffle, to anything other than 1000. The corrected line should read something of the form:

```
“shift=randi([-90*sr 90*sr],[ numofshuffle,1]); % -+ 90 sec shift”
```

Please update the code release at your earliest convenience.

Reviewer #1 (Remarks on code availability):

No concerns

Reviewer #2 (Remarks to the Author):

I thank the authors for the care of their response. I particularly appreciate the addition of FigureS3, which makes it clear that places were sampled from multiple trajectories.

the methods have overall been much clarified, and I consider that the paper is very strong in its demonstration, and should be published.

Reviewer #3 (Remarks to the Author):

The authors have performed a careful revision taking into account the points raised by the reviewers, and I do not wish to raise further points. This is an interesting paper.

We thank the editor and the reviewers for their useful comments and suggestions. Below is our answer to the comments of Reviewer #1.

Reviewer #1 (Remarks to the Author):

Piza et al. successfully addressed the overwhelming majority of my concerns, and the manuscript as a whole is greatly improved. I look forward to its eventual publication of this exciting work. I have only a few remaining comments and suggestions that should be readily addressable.

We once again thank the reviewer for the careful and thorough evaluation of the revised manuscript. We are particularly grateful for their assistance in improving the description and technical validity of our methods, as well as for providing critical perspectives and context to the readers. We believe that these contributions have greatly improved the manuscript. We have also corrected the typos and incorporated further clarifications based on their suggestions.

While I agree it is unfortunate that the original authors did not test for lack of Theta phase resetting in rats during head movements, the authors do have the data from Buzsáki's lab for rapid head movements in rats. The main issue is that the Authors cannot determine which head movements are orienting towards a visual target versus which head movements are re-allocation of the whiskers/nose. The Authors can at least average over all sufficiently fast head movements and look to see if you see any clear Theta phase resetting in the available rat data with an appropriate caveat how results may be affected by the consideration of all head movements. For example, the Authors can simply make Fig. S8c-e for the rat data and report how consideration of head movements not directed towards visual targets could be biasing the results towards the non-appearance of Theta phase resetting in rats. I understand that this analysis will be limited in the sense that it will not be able to distinguish between head movements that are and are not directed towards a visual target, but the analogous. Simply make this caveat clear to the reader and show the results, whatever they may be.

The reviewer has noticed we are conservative at using the neural data from the rat to analyze theta oscillations and their relationship to phase resetting. In the main text's discussion regarding head/gaze phase resetting, we note that non-human primate studies have generally shown that theta oscillations don't occur as prominently and rhythmically in awake monkeys as they do in rats, but instead occur in rather short bouts (Buzsáki, 2002; Courellis et al., 2019; Doucet et al., 2020; Jutras et al., 2013; Mao, 2022; Mao et al., 2021; Winson, 1974). We propose that certain events such as rapid head movements, generate a corollary discharge signal that reaches the hippocampus, which might serve as a synchronization signal aligning the phase of the LFP and aiding encoding based on imminent sensory inputs (Katz et al., 2022). This can be manifested as short theta bouts in the LFP (see **Figure 8a** and line 599 of the main text).

Additionally, our statement in line 432 of the introduction: "These differences may have impacted the physiological mechanisms of spatial navigation and specializations in the hippocampus of the two species" was intended to serve as a hypothesis contextualizing the main results to follow. Our primary findings contradict the mainstream notion from rodent literature that hippocampal place cells are predominantly allocentric and lack significant head direction encoding (Muller et al., 1994; O'Keefe & Nadel, 1978). We also report abundant angular head velocity encoding in hippocampal neurons, which is not reported in rodent hippocampal literature (see line 512 of the main text).

Our results do not rule out the possibility of theta phase resetting in the rat hippocampus in response to head movements. A similar efference copy-related modulation may occur in the hippocampus of the rat, for example, aligned to whisker movement or sniffing, which could happen when the animals move the head. To disentangle these possibilities from head movement one requires a particular design. We discussed this issue many times and we do not feel comfortable deriving conclusions from the data we downloaded since it does not contain information about these events. Nonetheless, to satisfy the reviewer curiosity and ours we conducted the same phase clustering analysis (**Figure S8c-e** of the main text) with rat hippocampal data (CRCNS.org, Data sharing, hc-2 data set) we observed a significant modulation of the average LFP signal aligned to head movement onset (**Response figure 1a**) and significant phase clustering (**Response figure 1b-d**) for delta (0.5-4Hz) band-pass filtered LFP but not theta (4-10Hz) band-pass filtered LFP (**Response figure 1e-g**).

We are enthusiastic but we believe that these effects need additional experimental validation and more thorough investigation than feasible within the current manuscript's scope. Consequently, we have chosen not to include these new results in the main manuscript. We are actively working on a new manuscript where we expand on the phase resetting effects, we see in our data to other motor behaviors. We are also collaborating with a lab that conduct studies in rats to specifically test this question under a better controlled experimental condition. We thank the reviewer for this suggestion since it has opened a new avenue of investigation for us, and we hope the reviewer understands our position.

Response figure 1 (a) Head movement triggered average LFP in the rat hippocampus, the shaded area corresponds to the 95% confidence interval. Statistical significance was evaluated by comparing the LFP raw values at times -120ms and +120ms using a two-sided Wilcoxon signed-rank test ($p=5.019e-13$, $Z=7.22$) **(b)** heatmap of mean delta (0.5-4Hz) band-pass filtered phase concentration around a 300ms window centered on head movement onset dotted gray line indicates -120ms, dotted purple line indicates +120ms **(c)** phase distribution during -120ms (gray) and +120ms (purple) time epochs. Green and red lines indicate circular mean and vector length of the gray and purple distribution respectively. **(d)** Mean Rayleigh's Z statistic for non-uniformity of circular data corresponding for a 300ms window centered on head movement onset. The pink dotted line indicates the critical value for significance at $\alpha=0.01$. **(e,f,g)** Corresponds to the same plots as in **(b,c,d)** respectively, computed for theta (4-10Hz) band-pass filtered LFP.

For lines 691-692 on page 23, “We observed modulations of both LFP and firing rate ~120ms before the head peak velocity (Fig. 8a-c)”. I have a concern regarding the significant activity you observed in the LFP and firing rate traces at time points <120 ms before peak head velocity. I suspect it is due to modulation from the previous jump in head-gaze direction. Can you show this significant activity still exists when you exclude jumps in head-gaze direction that begin within 120 milliseconds of the end of another jump in head-gaze direction?

This is a very important observation. We did compute the inter-event interval and found that less than 1% of all detected head movements happened between 120ms from each other, and they occur with a mean inter-event interval of roughly 200-250ms. Additionally, we chose to align the average LFP responses to the peak velocity of the head movement since it is a clearer signal than head movement onset, which at lower speed movements can be too close to the noise floor. By aligning the LFP signal to the peak velocity, the start of the modulation appears to occur early at approximately the onset of the head movement. This effect is clear if we plot the average LFP signal aligned to head movement onset (see **Response figure 2** below).

Response figure 2 Head movement triggered average LFP aligned to head movement onset for Subject C **(a)** and Subject P **(b)**, the shaded area corresponds to the 95% confidence interval. Dashed line corresponds to time 0. Observe the alignment affected the shape of the signal, as anticipated, but not the frequency.

For lines 832-834 on page 27, “Quaternion constructs avoid gimbal lock by representing space as a three-dimensional sphere in four-dimensional space where any loop along its topology can be continuously contracted to a single point”. This is an excellent addition. That being said, you have a typo (“it's topology” \diamond “its topology”). Furthermore, I think it would benefit the reader to include some more physical intuition here. For example, you could add how equivalently, rotations in three-dimensional space are simply connected. On a separate note, were the real components of your quaternionic representations nonzero? The most common way to represent rotations with quaternions are purely imaginary, with the real q_0 term reserved for the exponent of scale transformations. Hence, $q_0=0$ corresponds to the trivial transformation of scale. If indeed $q_0=0$, it would be good to mention that to the reader, as well, along with what it means in terms of trivial scale transformations.

Thank you for pointing this out, we have now corrected the typo and clarified further in the main text (see line 835) “Quaternion constructs avoid gimbal lock by representing space as a three-dimensional sphere in four-dimensional space where any loop along its topology can be continuously contracted to a single point¹⁵⁶, that is, any two unit quaternions representing rotations can be connected by a continuous path within this space and are considered simply connected”. The quaternion representation obtained from the tracking software (Motive, Optitrack) are normalized to unit length. We have also added a reference that can be useful to readers that are interested in expanding upon the representation of rotations using quaternions (Hatcher, 2002).

On lines 847-848 on page 27, “Movements with speeds higher than 300cm/s were deemed artifactual [sic.]”. You are missing a period here.

Thank you for catching this, we have corrected this.

Of lesser importance, I noticed a minor error in your code release. The error is on line 143 of ‘3Dmarms/Speed Score/Speed_Score_demo.m’. The problem only manifests if you change the number of shuffles, numofshuffle, to anything other than 1000. The corrected line should read something of the form:

```
“shift=randi([-90*sr 90*sr],[ numofshuffle,1]); %-+ 90 sec shift”
```

Please update the code release at your earliest convenience.

Thank you for this observation, we have updated the code release.

References

Buzsáki, G. (2002). Theta oscillations in the hippocampus. *Neuron*, 33(3), 325–340.

[https://doi.org/10.1016/s0896-6273\(02\)00586-x](https://doi.org/10.1016/s0896-6273(02)00586-x)

- Courellis, H. S., Nummela, S. U., Metke, M., Diehl, G. W., Bussell, R., Cauwenberghs, G., & Miller, C. T. (2019). Spatial encoding in primate hippocampus during free navigation. *PLOS Biology*, *17*(12), e3000546. <https://doi.org/10.1371/journal.pbio.3000546>
- Doucet, G., Gulli, R. A., Corrigan, B. W., Duong, L. R., & Martinez-Trujillo, J. C. (2020). Modulation of local field potentials and neuronal activity in primate hippocampus during saccades. *Hippocampus*, *30*(3), 192–209. <https://doi.org/10.1002/hipo.23140>
- Hatcher, A. (2002). *Algebraic Topology*. Cambridge University Press.
- Jutras, M. J., Fries, P., & Buffalo, E. A. (2013). Oscillatory activity in the monkey hippocampus during visual exploration and memory formation. *Proceedings of the National Academy of Sciences*, *110*(32), 13144–13149. <https://doi.org/10.1073/pnas.1302351110>
- Katz, C. N., Schjetnan, A. G. P., Patel, K., Barkley, V., Hoffman, K. L., Kalia, S. K., Duncan, K. D., & Valiante, T. A. (2022). A corollary discharge mediates saccade-related inhibition of single units in mnemonic structures of the human brain. *Current Biology*, *32*(14), 3082-3094.e4. <https://doi.org/10.1016/j.cub.2022.06.015>
- Mao, D. (2022). Neural Correlates of Spatial Navigation in Primate Hippocampus. *Neuroscience Bulletin*, *39*(2), 315–327. <https://doi.org/10.1007/s12264-022-00968-w>
- Mao, D., Avila, E., Caziot, B., Laurens, J., Dickman, J. D., & Angelaki, D. E. (2021). Spatial modulation of hippocampal activity in freely moving macaques. *Neuron*, *109*(21), 3521-3534.e6. <https://doi.org/10.1016/j.neuron.2021.09.032>
- Muller, R. U., Bostock, E., Taube, J. S., & Kubie, J. L. (1994). On the directional firing properties of hippocampal place cells. *Journal of Neuroscience*, *14*(12), 7235–7251. <https://doi.org/10.1523/JNEUROSCI.14-12-07235.1994>
- O'Keefe, J., & Nadel, L. (1978). The Hippocampus as a Cognitive Map. In *Oxford University Press: Oxford, UK*. (1978). Oxford University Press. <http://www.cognitivemap.net/>

Winson, J. (1974). Patterns of hippocampal theta rhythm in the freely moving rat.

Electroencephalography and Clinical Neurophysiology, 36, 291–301.

[https://doi.org/10.1016/0013-4694\(74\)90171-0](https://doi.org/10.1016/0013-4694(74)90171-0)